# Impact of the COVID-19 pandemic on incident diagnoses in German refugee centres 2018 to 2023

Kayvan Bozorgmehr [1,2,7] ✉, Stella Erdmann[3,7], Sven Rohleder[1], Consortium Pri.CareNet* & Rosa Jahn[2]

The COVID-19 pandemic may have affected morbidity patterns of residents in refugee centres, but empirical evidence is scarce. We utilised linked data from a health surveillance network in refugee centres of three German federal states, employing a quasi-experimental design to examine the effects of the COVID-19 pandemic on newly diagnosed medical conditions. Doctors coded routine diagnoses on-site in healthcare facilities for refugee patients. Our analysis encompasses the timeframe from October 2018 to April 2023 and includes individual-level data for 109,175 refugees. This data resulted in 76,289 patient-months across 21 refugee centres, with a total occupancy of 144,012 person-months. Here, we employ segmented regression analyses, adjusting for time trends, socio-demographic factors, occupancy, and centre characteristics, to evaluate the COVID-19 pandemic's impact on incident diagnosis patterns among refugees. We show how the COVID-19 pandemic altered diagnosis patterns among refugees in German centres. Notably, incidents of injuries, mental disorders, psychotherapeutic drug prescriptions, and genitourinary diseases rose, while respiratory diseases decreased, later rebounding. An increase in injury-related diagnoses suggests heightened violence experiences during flight or in centres. Mental disorder diagnoses and psychotherapeutic drug prescriptions rose, reflecting pandemic-related stressors and the pandemic's multifaceted impact on refugee health.

The COVID-19 pandemic and related responses have impeded progress towards the Sustainable Development Goals worldwide[1]. The pandemic revealed and exacerbated pre-existing inequalities, with migrants and refugees being particularly affected by multiple adverse conditions, which put them at higher risk of suffering from COVID-19 or from the consequences of pandemic control measures[2,3]. Despite early appeals by the World Health Organization to adequately consider migrants and refugees in pandemic response plans[4], many countries side-lined migrants and refugees in their global health response[5-7].

Refugees in camps have been at particular risk of acquiring COVID-19 due to poor hygienic conditions and over-crowding[4,8-10]. They may face additional health disadvantages during pandemics due to their disintegration into the regular health care system, as well as language barriers that may hamper their access to necessary health information. Additionally, many health and immigration authorities responded to the COVID-19 pandemic with non-evidence-based measures, including mass quarantine, which refers to indiscriminate movement restrictions for all camp inhabitants and bans of in-and-out

[1]Department of Population Medicine & Health Services Research, School of Public Health, Bielefeld University, Bielefeld, Germany. [2]Section Health Equity Studies & Migration, Heidelberg University Hospital, Heidelberg, Germany. [3]Institute of Medical Biometry, Heidelberg University, Heidelberg, Germany. [7]These authors contributed equally: Kayvan Bozorgmehr, Stella Erdmann. *A list of authors and their affiliations appears at the end of the paper. ✉ e-mail: kayvan.bozorgmehr@med.uni-heidelberg.de

movements. Such measures have not only put refugees at higher risk of COVID-19 infection[10], but also led to disruptions of essential health and social services[11], which in turn further increased the pre-existing isolation and marginalisation of refugees. Early guidance of the European Centre for Disease Prevention and Control stated that mass quarantine of refugee centres should be avoided due to the potential negative impact on refugees' mental health[7]. Nevertheless, such practices were widespread. In Germany, for example, mass quarantine was not only used in the early pandemic phases but continued into 2022[10]. Combined with other pandemic control and mitigation measures[7], life in refugee centres became even more regulated and subject to external control of authorities, severely affecting and interfering with refugees' autonomy.

In Germany, pandemic measures for refugees involved compulsory PCR testing upon arrival, followed by mandatory quarantine, additional testing, and pre-emptive quarantine after transfer from one reception centre to another. Measures included social distancing, mandatory mask-wearing, movement restrictions, and the reduction or suspension of on-site services and activities, such as leisure and shared spaces, to control and mitigate the spread of the pandemic[11].

Similar measures, including testing, individual and mass quarantine, and social distancing, were used in COVID-19 and other infectious disease containment efforts in refugee camps beyond the German context as well[12]. While such measures are generally intended to protect refugees and prevent disease transmission, refugees may experience them as coercive[13]. This is exacerbated by often insufficient communication and coordination strategies of health and immigration authorities in charge of the centres[11,14]. Furthermore, the COVID-19 pandemic led to disruptions of administrative processes of the asylum system, which in turn amplified uncertainty for refugees with precarious residence status or whose decision on their asylum application was pending.

The COVID-19 pandemic and its related responses, combined with disruptions of health and social services, had the potential to function as severe stressor for refugees in institutionalised settings such as camps or refugee centres. However, evidence on the impact of the COVID-19 pandemic on refugee populations is rare[15]. The lack of evidence has been attributed to pre-existing weaknesses of health monitoring systems[16], which often fail—even in high-resource countries—to capture health of refugees in a manner that is both comprehensive and comparable across time and space[17]. Systematic reviews from early[2,3] and later stages of the pandemic[15], as well as preliminary results[18] of ongoing systematic reviews[19], indicate that there is a paucity of quantitative evidence on health impacts of the COVID-19 pandemic in refugee populations.

We used data from a novel multi-centre health surveillance network in refugee centres[20] of three German federal states to analyse the impact of the COVID-19 pandemic on incident diagnosis patterns among refugees between 2018 and 2023.

We conceived the onset of the pandemic as a quasi-experimental situation, i.e., a non-randomized intervention design in which the assignment of individuals or populations to exposed or controls occurs quasi-randomly and is out of control of study participants and researchers alike. The design resembles an interrupted time series design[21], to perform a longitudinal analysis of changes in the incidence proportion of coded medical diagnoses among refugees seeking care in on-site healthcare facilities in refugee centres by using segmented regression models (details on models see below, results and methods). We thereby considered time trends before and during the pandemic (referred to as pre- and peri-pandemic time trends), key socio-demographic characteristics of refugees, number of inhabitants in, as well as variations stemming from characteristics of refugee centres. We used data from an electronic health records (EHR) software, which is used by on-site healthcare providers across refugee centres in the network for the structured and standardised recording of diagnoses,

symptoms, prescriptions and follow-up procedures. We used these individual-level data to determine the incidence of newly diagnosed conditions, and linked this information with aggregated data on the occupancy of refugee centres (i.e., number, age, and sex of inhabitants of refugee centres) provided by immigration authorities to obtain reliable denominators (see methods). The data covers individual-level data of 109,175 refugees (unique individuals), who visited on-site healthcare facilities at least once and of which we derived an analysis set comprising 76,289 patient-months from 21 refugee centres. The centres are located in three German federal states and comprised a total occupancy of 144,012 person-months of refugees during the observation period. We used March 2020 as break point in a segmented regression analysis to investigate the impact of the COVID-19 pandemic on a total of 21 indicators reflecting newly coded diagnoses on non-communicable diseases, physical conditions, mental conditions, and infectious diseases. Indicators were constructed using diagnosis categories based on International Classification of Diseases (ICD-10-GM Version 2021) and drug prescriptions based on the Anatomic Therapeutic Classification (ATC 2023) from the EHR. Further subsets of the available data sources were used to perform sensitivity analyses (details see methods).

In this work, we show that the COVID-19 pandemic significantly altered diagnosis patterns among refugees in the analysed centres. Incidents of injuries, mental disorders, psychotherapeutic drug prescriptions, and genitourinary diseases rose, while respiratory diseases decreased initially, but increased again in later stages of the pandemic. The identified pattern provides evidence of the detrimental impact of the COVID-19 pandemic on health, reflected by a rise in potential consequences of violence (and other external causes) and mental health problems.

## Results

### Descriptive results

The main analysis period spanned 56 months (October 2018–April 2023) across up to 21 refugee centres, which were successively included in the surveillance network, yielding a total of 314 centre-months of observations on coded diagnoses. About one third of the observations (36%) stem from central registration- and reception-centres, in which refugees are placed for several weeks to a few months. About two thirds (64%) was derived from peripheral reception centres, to which refugees are transferred until a decision is made on their asylum application, and in which they stay up to 18 months or longer. Of the total occupancy of these centres (144,012 person-months), on average 64% were male and 75% adults (aged 18 years or above) (Table 1). The five most frequent countries of origin among all refugee patients ($N = 215,864$) across all centres (2018–2023), weighted by the centre's population size, were Afghanistan (Mean ± standard deviation: $18\% \pm 18$), Iraq ($3.2\% \pm 4.8$), Nigeria ($8.6\% \pm 13$), Syria ($12\% \pm 19$) and Türkiye ($4,2\% \pm 7.6$), while percentages broken down by year varied (Supplementary File, Table S1). Patients were, on average, about one year younger after March 2020 compared to the average age before onset of the pandemic (see Supplementary File, Chapter 3.3, for a list of mean age by centre before and after onset of the pandemic).

The most frequently coded medical diagnoses groups were "Diseases of the respiratory system" (ICD J00-J99), followed by "Certain infectious and parasitic diseases" (ICD A00-B99)" and "Diseases of the digestive system" (ICD K00-K93) with an incidence proportion of 6.0%, 4.8%, and 4.7%, respectively. These were followed by coded diagnoses on "Mental and behavioural disorders" (ICD F00-F99) with a (weighted) mean average incidence proportion of 4.0%. The mean (weighted) incidence proportion was 0.77% for prescriptions of psychoactive drugs (Fig. 1). The least frequently coded medical diagnosis groups were "Diseases of the blood or blood-forming organs and certain disorders involving the immune mechanism" (ICD D50-D90) and "Neoplasms" (ICD C00-D48) with a (weighted) average incidence

**Table 1 | Weighted sociodemographic characteristics of occupancy in refugee centres, 2018–2023, *N* = 144,012 person-months**

| Centre-months | 2018 | 2019 | 2020 | 2021 | 2022 | 2023 | Total |
| --- | --- | --- | --- | --- | --- | --- | --- |
|  | 6 | 14 | 9 | 11 | 12 | 10 | 62 |
| Sociodemographic characteristics | | | | | | | |
| % Male | | | | | | | |
| Mean ± sd | 70 ± 7.6 | 63 ± 15 | 63 ± 15 | 61 ± 8.4 | 61 ± 12 | 68 ± 17 | 64 ± 13 |
| Median (Q1, Q3) | 70 (63, 73) | 66 (55, 76) | 66 (49, 71) | 64 (50, 68) | 66 (53, 68) | 72 (66, 81) | 66 (58, 72) |
| Min–max | 60–81 | 34–82 | 40–87 | 48–71 | 37–74 | 36–87 | 34–87 |
| CI | [62,78] | [54,71] | [51,74] | [56,67] | [53,68] | [55,80] | [60,67] |
| W | 67 ± 6.7 | 64 ± 11 | 64 ± 11 | 62 ± 8 | 65 ± 9.2 | 74 ± 12 | 64 ± 3.6 |
| % Adults | | | | | | | |
| Mean ± sd | 79 ± 6.2 | 78 ± 6.2 | 76 ± 12 | 71 ± 8 | 70 ± 8.9 | 79 ± 9 | 75 ± 9 |
| Median (Q1, Q3) | 80 (76, 84) | 77 (73, 82) | 80 (68, 82) | 72 (70, 78) | 72 (69, 75) | 81 (73, 88) | 76 (70, 81) |
| Min–max | 70–86 | 68–89 | 58–94 | 53–81 | 50–81 | 65–90 | 50–94 |
| CI | [73,86] | [74,81] | [67,86] | [66,77] | [65,76] | [73,86] | [73,78] |
| W | 78 ± 5.5 | 77 ± 6.1 | 77 ± 8.6 | 72 ± 6.7 | 73 ± 7.6 | 81 ± 9.1 | 76 ± 3.9 |
| Average monthly number of inhabitants (occupancy) | | | | | | | |
| Mean ± sd | 409 ± 374 | 306 ± 260 | 350 ± 257 | 323 ± 257 | 510 ± 539 | 529 ± 439 | 401 ± 367 |
| Median (Q1, Q3) | 268 (224, 413) | 256 (133, 377) | 224 (156, 539) | 228 (130, 520) | 319 (246, 514) | 328 (265, 768) | 269 (175, 520) |
| Min–max | 131–1147 | 41–1058 | 132–884 | 62–903 | 201–2102 | 124–1540 | 41–2102 |
| CI | [16, 801] | [156, 456] | [153, 548] | [151, 496] | [168, 853] | [215, 843] | [308, 494] |
| W | 409 ± 374 | 306 ± 260 | 350 ± 257 | 323 ± 257 | 510 ± 539 | 529 ± 439 | 405 ± 96 |

*SD* Standard deviation, Q1, Q3: first and third quartile, min: minimum, max; maximum and CI: 95% Confidence interval of the "weighted mean facility observation" values (weighted by *n*$_{occ}$); W 2018_2023: "weighted annual" mean ± standard deviation (weighted by the mean occupancy of a facility in the respective year); W total: mean value ± standard deviation of weighted annual mean values.

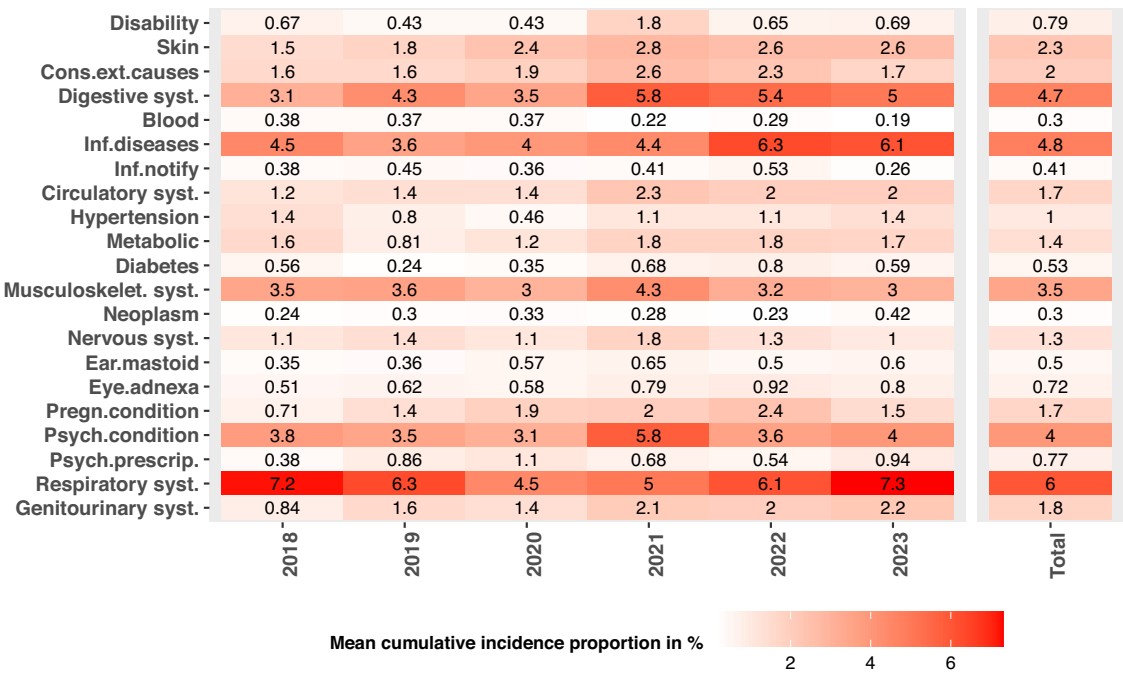

**Fig. 1 | Heat map of weighted mean cumulative incidence proportions of 21 indicators, 2018–2023.** Cumulative incidence proportions are weighted by the mean occupancy of a facility in the respective year or time period (2018–2023). Detailed values of weights as well as standard deviations and 95% confidence intervals are listed in the Supplementary File. Indicator definitions are based on diagnoses (ICD-10 Codes) and prescriptions (ATC-Codes), recorded in the electronic health records. *N* = 144,012 person-months. For definition of short indicator labels, see Table 2. Source data are provided in the 'source_data.xlsx' file.

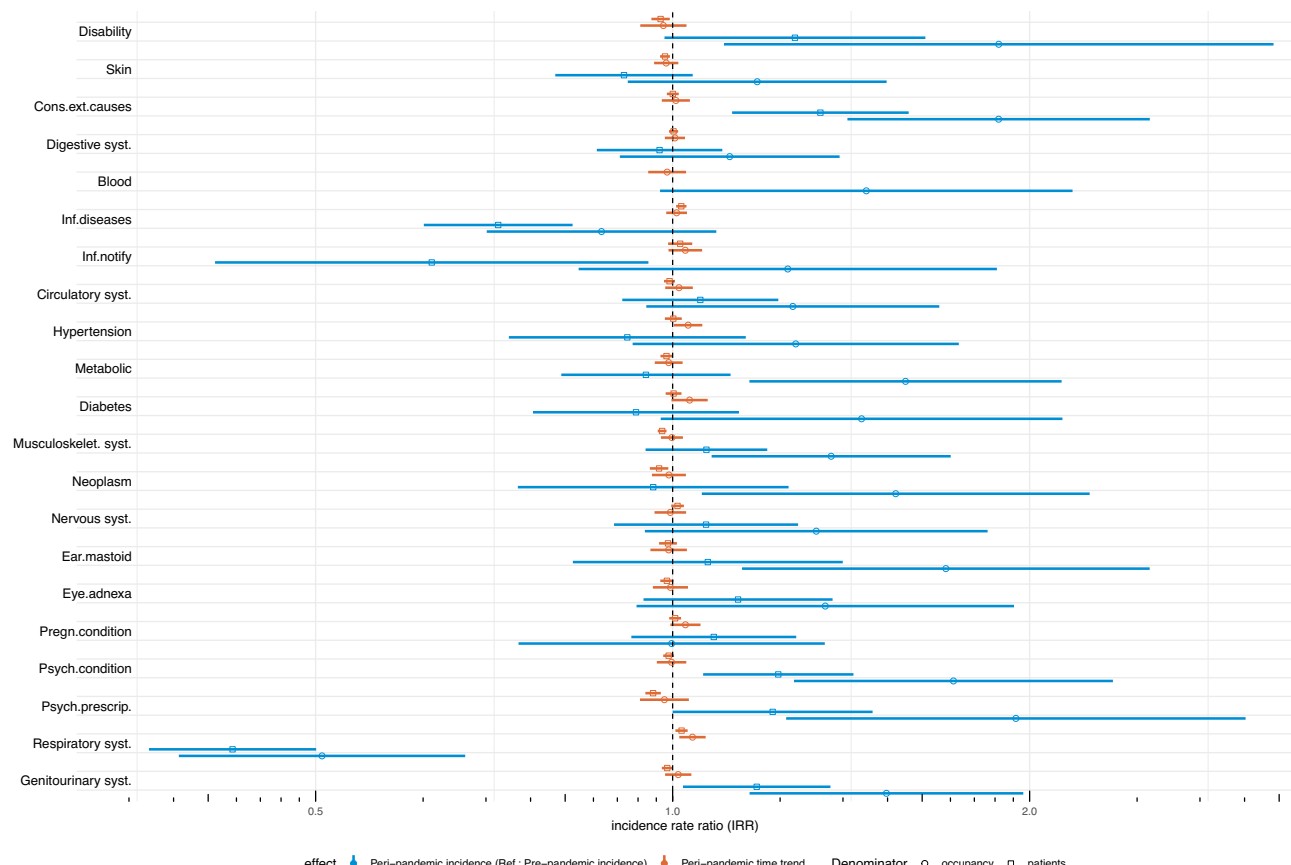

**Fig. 2 | Impact of the COVID-19 pandemic on 21 indicators, adjusted incidence rate ratios (IRR), 2018–2023.** The incidence rate ratios (IRR) are presented with associated 95% confidence intervals (CI). Estimates are derived for each indicator separately and adjusted for the proportion of males, the proportion of adults, secular trends, and potential influences of the characteristics of refugee centres (random intercept). Y-axis: log-scale. Circles: Estimates derived from data subset 2 using monthly occupancy numbers in refugee centres as denominator (main analysis). Squares: Estimates derived from data subset 1 using monthly patient numbers of clinics in refugee centres as denominator and additionally adjusting for country of origin (Sensitivity analysis 1). Note, that the model for the variable "Diseases of the blood and blood-forming organs and certain disorders involving the immune mechanism" (short label: Blood) did not converge in Sensitivity analysis 1. Therefore, the results of this variable are given for the main analysis only. For definition of short indicator labels, see Table 2. Source data are provided in the 'source_data.xlsx' file.

proportion of 0.3%, respectively. Notifiable infectious diseases (except COVID-19; and for example, comprising measles, chicken-pox, hepatitis, or HIV) according to section 36 of the German Infection Protection Act (IfSG) were also found among the least frequently coded indicators with an average (weighted) incidence proportion of 0.41% (Fig. 1).

Means, standard deviations, median, minimum, and maximum as well as 95% confidence intervals (CI) of the weighted cumulative incidences of all 21 indicators stratified by year can be found in the Supplementary File, Table S1.

### Impact of the COVID-19 pandemic

The impact of the COVID-19 pandemic on the 21 health indicators can be found in Fig. 2. The statistical model we used (mixed-effects zero-inflated negative binomial models) was fitted to investigate the average difference in incidence proportions between time points during (*peri*-pandemic) and before (*pre*-pandemic) the onset of the pandemic as incidence rate ratios (IRR), while adjusting for the proportion of males and adults, secular trends (time as discrete variable), as well as potential influences of the characteristics of refugee centres, which we considered as random intercept. The models account for the fact that many individuals in the refugee population have never experienced any of the events measured by the 21 health indicators, and as such will have zeros coded in the data, at least in major time periods (therefore: zero-inflated model). These "zeros" are treated as real and informative

data, as opposed to other models that simply ignore and exclude these non-events from the analysis. The mixed-effects models also allow to account for differences in the dependent variables that are due to differences between groups or contexts, like refugees from different reception centres or states, rather than attributable to differential exposure to the independent variable (i.e., the pandemic). The impact was additionally evaluated by the mutually adjusted *peri*-pandemic time trend. The exact values of the IRR with 95% CI, *p*-values as well as the results of the fitted models, can be found in Supplementary Chapter 1. Sensitivity analyses comprised further adjustment for differences within and between centres in underlying countries of origin over time, but this required using patients (instead of occupancy) as denominators as the information was available in the EHR but not in occupancy data (Supplementary Chapter 2).

The adjusted IRRs of conditions related to "Disabilities" (IRR: 1.88 [1.1–3.21]), "Injury, poisoning and certain other consequences of external causes (S00–T98)" (IRR: 1.88 [1.4–2.53]), "Mental and behavioural disorders (F00–F99)" (IRR: 1.73 [1.27–2.35]), "Prescription rates of psychopharmaceutic drugs" (IRR: 1.95 [1.25–3.04]) as well as "Diseases of the genitourinary system (N00–N99)" (IRR: 1.51 [1.16–1.98]) were significantly higher in the peri-pandemic compared to the pre-pandemic time period. The IRRs remained elevated or at least marginally significant with lower point-estimates in sensitivity analyses which were further adjusted for countries of origin (Fig. 2). The adjusted incidence of "Diseases of the respiratory system (J00–J99)"

significantly declined (IRR: 0.51 [0.38–0.67]) in the peri-pandemic period, and even more under further adjustment for countries of origin (Fig. 2). Despite the use of different denominators for the calculation of incidence proportions, and different adjustment variables, results for these indicators were consistent.

Several indicators showed significant increases (IRR and 95% CI > 1.0) in the peri-pandemic period when using occupancy numbers as denominators, but the effects disappeared in the sensitivity analyses under adjustment for countries of origin and usage of patient numbers as denominator for calculating incidences (Fig. 2). Among these were "Endocrine, nutritional and metabolic diseases (E00-E90)", "Diseases of the musculoskeletal system and connective tissue (M00-M99)", "Neoplasms (C00-D48)", and "Diseases of the ear and mastoid process (H60-H95)". The results for these indicators were less consistent and amenable to variation after adjustment for countries of origin.

The incidence of "Certain infectious and parasitic diseases (A00-B99)" was non-significant when using occupancy numbers as denominator (IRR: 0.87 [0.7–1.09]), but showed a significant decline during the pandemic compared to pre-pandemic time periods when using patients as denominator and adjusting for countries of origin (IRR: 0.63 [0.41–0.95]). A similar pattern was observed for notifiable infectious diseases according to German national law (IfSG), for which the incidence was unaffected by the pandemic when using occupancy as denominator (IRR: 1.25 [0.83–1.88]), but turned significant with IRR < 1.0 in the sensitivity analysis with patients as denominator and further adjustment for countries of origin (Fig. 2).

No impact of the COVID-19 pandemic was found on all other indicators, which consistently showed non-significant results both in main and sensitivity analyses (Fig. 2).

Results of further sensitivity analyses using other data subsets with different inclusion or selection criteria confirmed the patterns described above, and did not show substantially different model estimates (Supplementary Chapter 2).

The peri-pandemic time trends were non-significant or only marginally significant for 17 of the 21 indicators in the main analysis (Fig. 2 and Supplementary Chapter 1.1), indicating no major time trends beyond the mutually adjusted average impact of the pandemic. Only two indicators (Respiratory conditions, and Hypertension) showed significant peri-pandemic time trends *over and above* the average impact and the effects of co-variables: For each additional month in the peri-pandemic time period, the incidence of "Hypertensive disorders (I10-I15)" increased by 3%. "Diabetes Mellitus (E10-E14)" and conditions related to "Pregnancy, childbirth and the puerperium (O00-O99)" were marginally significant. Adjusting for country of origin (in sensitivity analyses with patients as denominator) resulted in non-significant time trends. While the average impact of the pandemic led to a decline in "Diseases of the respiratory system (J00-J99)", the peri-pandemic time trends increased by 4% per each additional month in the peri-pandemic period (Fig. 2 and Supplementary Chapter 1.1). In the sensitivity analysis, patterns remained unaffected when adjusting for country of origin.

The results of Sensitivity analysis 4, which used a nuanced approach to the peri-pandemic phase, are presented in Supplementary File, Chapter 5 (Fig. S26). The analysis shows that the results of the main analysis are widely robust to the nuances of different pandemic phases, as the observed patterns for e.g., disability, consequences of external causes of injuries, circulatory conditions, and psychological conditions remain unaffected and are consistent to patterns in our main analysis. Estimates for respiratory conditions are also consistent with previous main and sensitivity analysis, showing tendencies to declining incidence in the early phases of the pandemic, and rising trends in later phases.

Results of Sensitivity analysis 5, which examined seasonality effects, are presented in Supplementary File, Chapter 5 (Fig. S27). The results provide evidence of seasonality effects, particularly for "consequences of external causes of diseases", with higher incidences in summer periods compared to spring, winter and autumn, respectively. This may be due to route dependent effects (more dangerous routes during summer), heat related effects on incident violence, or higher probability of injuries related to outdoor activities. The incidence of "psychological conditions" was higher in winter periods (compared to summer periods), while respiratory conditions were significantly higher in winter and autumn (compared to summer periods). No considerable seasonality effects were observed for the remainder of diagnoses. The patterns prove the plausibility of the surveillance data, and are overall consistent with the patterns observed in the main analysis.

### Counterfactual analysis of selected indicators

To further consolidate the findings of our analysis, we performed a counterfactual analysis by predicting the expected incidence proportions of analysed health indicators from the models, while considering the respective age- and sex-distribution of refugees in the occupancy. The counterfactual measures reflect the incidence proportions that would have been expected if the COVID-19 pandemic had not happened. We visualised the expected versus the observed incidence as well as estimated counterfactual incidence of selected outcomes for which the above-mentioned estimates (for both the main analysis and sensitivity analysis) showed consistent results with either significantly elevated (Figs. 3–6) or reduced adjusted IRR as impact of the COVID-19 pandemic (Fig. 7).

### Discussion

The COVID-19 pandemic had a considerable impact on incident diagnosis patterns among refugees residing in refugee centres in Germany. Between October 2018 and April 2023, the incidence of "Injury, poisoning and certain other consequences of external causes", "Mental and behavioural disorders", "Prescriptions of psychotherapeutic drugs", and "Diseases of the genitourinary system" significantly and consistently increased, holding sociodemographic and centre-related factors constant. While "Diseases of the respiratory system" decreased in average, related diagnoses showed a rising trend in the later peri-pandemic time period after the initial decline.

The 88% rise in diagnoses related to injury and consequences of external causes may indicate an increased experience of violence before or during flight, e.g., due to the onset of the Russian war against Ukraine. However, this is unlikely as we adjusted for compositional effects, and because major nationalities among refugees remained widely constant 2018–2023, and Ukrainian nationals (whose numbers peaked from 2022 onwards) mostly resided in private accommodation outside reception centres. The results may also indicate higher violence during refugees' stay in the centres. Systematic reviews of media reports found that violent incidences in refugee centres during COVID-19 outbreaks were common. These were related to social tensions within the centres or to enforcement of the strict pandemic rules, which partially required, or were related to, police operations[10]. Previous research also showed that refugees living in reception centres are at a higher risk of experiencing mental health disorders due to crowded living conditions, reduced autonomy, lack of privacy and other adverse experiences often associated with life in such centres, such as uncertainty of asylum processes and legal status, geographical remoteness, and social isolation[22–26]. In addition, most reception centres do not provide psychosocial care services, making it difficult for patients to access appropriate care[27,28]. As a result, the estimated incidence of mental health disorders identified in our analysis (4% of patients) is likely to underestimate the true burden of mental health disorders. Our study also revealed a 73% increase in diagnoses related to mental disorders and a 95% increase in prescriptions of psychotherapeutic drugs following the onset of the pandemic. This suggests that the pandemic and related containment and mitigation

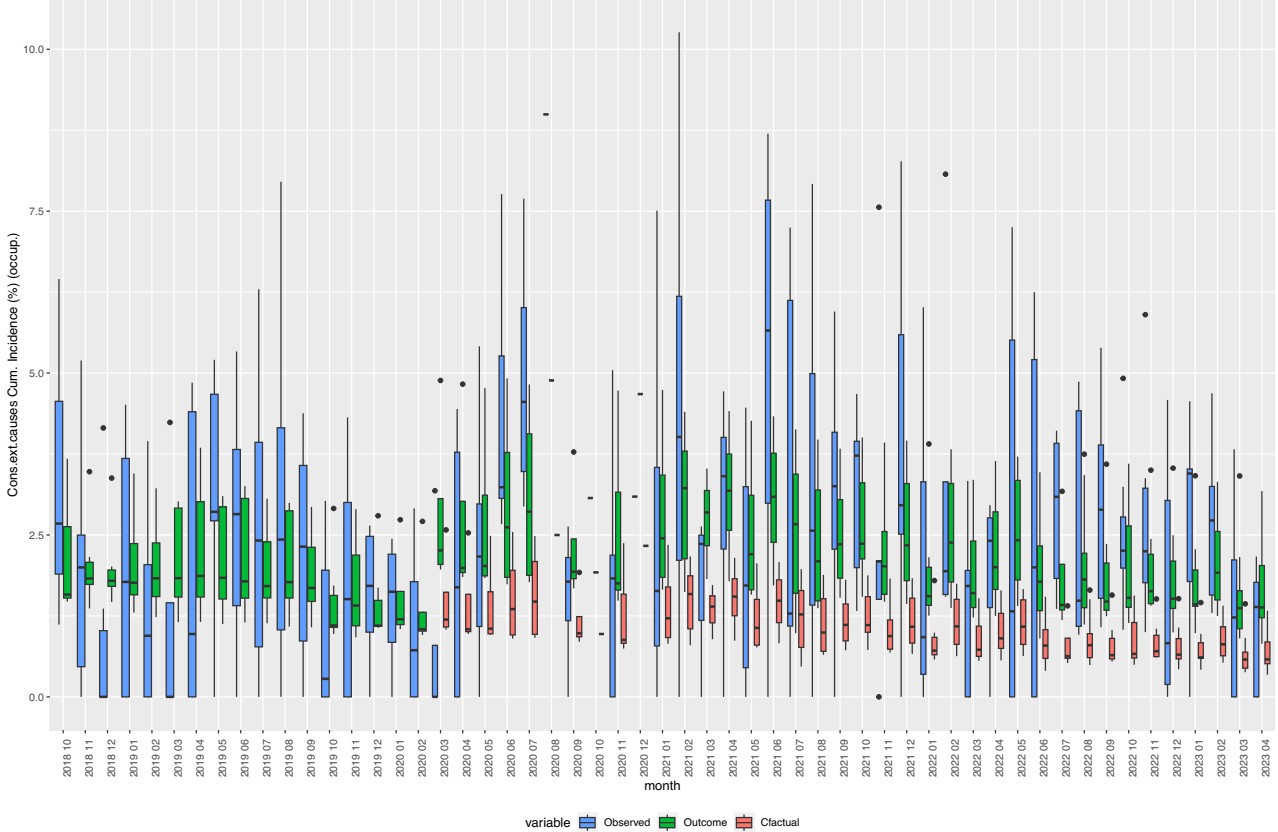

**Fig. 3 | Box Plots of observed, expected outcome, and expected counterfactual values of the incidence of "Injury, poisoning and certain other consequences of external causes (S00-T98)", 2018–2023.** Observed: observed crude incidence proportions, i.e., cases of individuals with one or more new diagnoses defined by the indicator variable divided by the total number of refugee centre inhabitants (occupancy) and multiplied by 100. Outcome: estimated outcome values based on negative binomial regression models, adjusted for age, sex, centre, and secular trends. Cfactual: estimated counterfactual values based on the negative binomial regression models, adjusted for age, sex, centre and secular trends while setting covid = 0. Cons.ext.causes: Injury, poisoning and certain other consequences of external causes. Cum. Incidence: Cumulative Incidence, occup.: occupancy. *Y*-axis: cumulative incidence in %. Boxes: interquartile range (IQR; the 25th and 75th percentiles). Whiskers: The upper whisker extends from the hinge to the largest value no further than 1.5 × IQR from the hinge. The lower whisker extends from the hinge to the smallest value at most 1.5 × IQR of the hinge. Data beyond the end of the whiskers are plotted individually by black dots. Horizontal black bar in boxes: Median. *N* = 836 centre-months. Source data are provided in the "source_data.xlsx" file.

measures have added to existing stressors, exacerbating the already challenging living conditions in refugee centres, and negatively impacting mental health.

Our analysis also showed a 51% increase in "Diseases of the genitourinary system" (N00-N99) following the onset of the COVID-19 pandemic, which includes, for example, urinary tract infections, kidney disease, and inflammatory or non-inflammatory diseases of male and female genital organs. This finding contrasts with available studies, which have found a reduction in genitourinary care utilisation and related diagnoses[29]. These were connected to general lockdowns but also to reductions in the scope, availability, and accessibility of genitourinary services often deprioritised in response to COVID-19[30]. In contrast, with very few exceptions, care provision in the on-site health facilities of the refugee centres included in this study has continued throughout the pandemic at roughly constant levels. The reduction in genitourinary service provision in surrounding hospitals may have increased on-site care-seeking among residents in refugee centres, potentially contributing to the increase in genitourinary conditions found in our data. However, as access to routine care outside of the centres is always low due to logistical, administrative, and language barriers, this effect is likely to be minor and other disease areas do not follow similar patterns. Another explanation may be that the increase in genitourinary conditions is due to early symptoms or complications[31,32] of infection with the Sars-CoV-2 virus. While our data

do not include individual-level information on COVID-19 infections (due to parallel reporting systems), we are aware of several outbreaks in some of the studied refugee centres, particularly during the early stages of the pandemic. Several literature reviews have shown that the virus can lead to wide range of urogenital complications, including acute kidney injury and lower urinary as well as genital infections[33–35]. The rise in diseases of the genitourinary system observed in this refugee population may also be explained by potential changes in sexual behaviour, including higher rates of transactional sex[36] and gender-based violence[37], and/or changes in access to sexual and reproductive health services outside of the facilities[38]. Further research is, however, required to confirm potential pathways.

Our study shows that respiratory conditions considerably declined during the pandemic, mostly likely as a consequence of the strict pandemic control measures implemented in the refugee centres. This finding is in line with international studies in refugee populations[39] and national studies in the resident population in Germany[40]. The rising time trend, on the other hand, may be an indication of the subsequent and sequential relaxation of measures over time with gradual, individual behaviour changes and lower adherence to and enforcement of social distancing or wearing of masks. Alongside these factors, waning immunity to other respiratory conditions may have contributed to the rebound. The rising time trend may also be an indication of the rather slow, and late implementation of

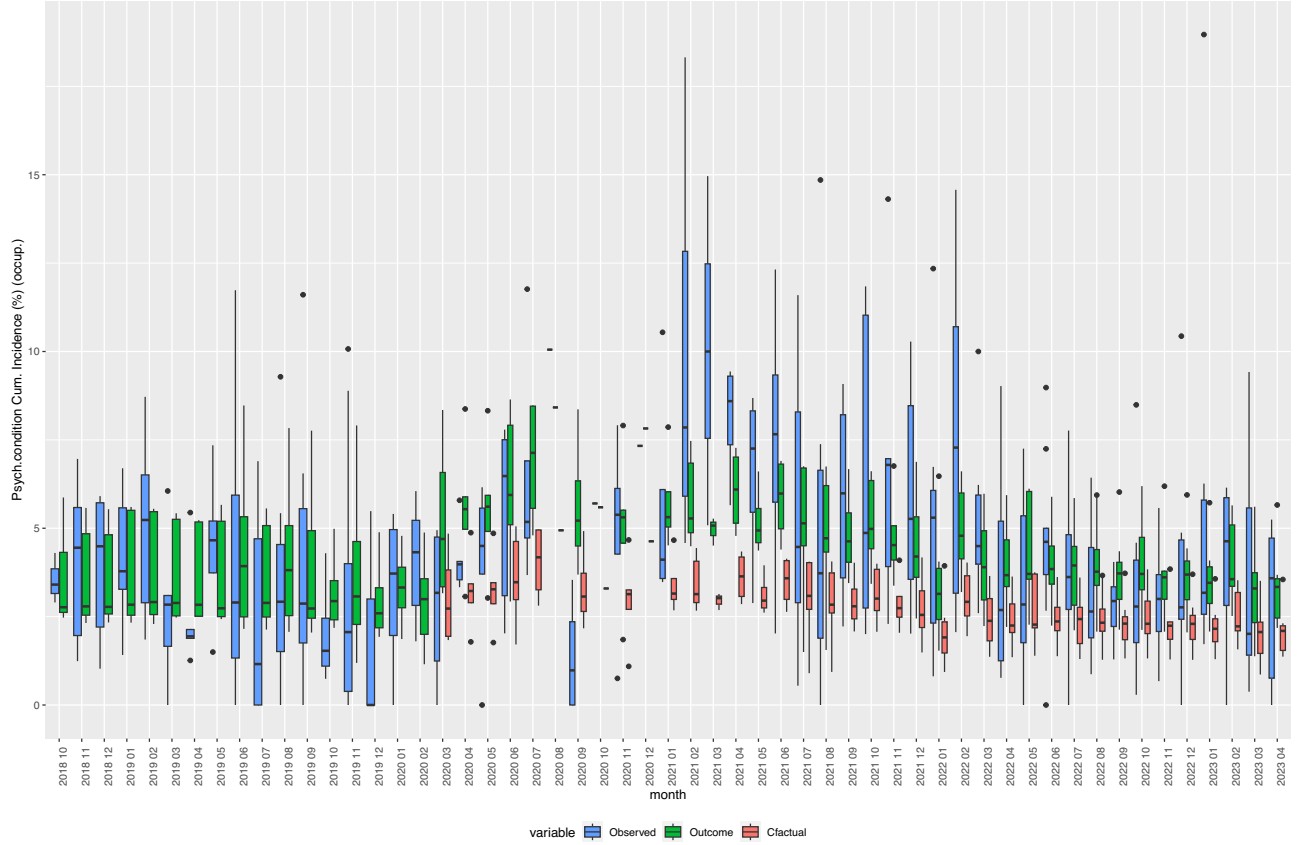

**Fig. 4 | Box Plots of observed, expected, and fitted values of the incidence of "Mental and behavioural disorders (F00-F99)", 2018–2023.** Cfactual: counter-factual (expected) values given the respective age- and sex-distribution of under-lying refugee population (occupancy) in reception centres at given time points. Fitted: fitted values based on negative binomial regression models, adjusted for age, sex, centre characteristics, and secular trends. Observed: crude incidence proportions, i.e., cases of individuals with one or more new diagnoses defined by the indicator variable divided by the total number of refugee centre inhabitants (occupancy). Psych. condition: Mental and behavioural disorders. Cum. Incidence: Cumulative Incidence, occup.: occupancy. Y-axis: cumulative incidence in %. Boxes: interquartile range. Whiskers: Range. Horizontal black bar in boxes: Median. N = 836 centre-months. Source data are provided in the 'source_data.xlsx' file.

effective vaccination measures against COVID-19. Media reports indicate and prove that despite the prioritisation by the national public health agency of refugees in the national vaccination policy, local implementation was ineffective and de-prioritised refugees in practice, leading to delayed vaccination up-to April 2021[41]. As formal evaluations of vaccination rates in refugee populations are scarce globally[15], no further insights exist to validate or assess the impact of potentially delayed vaccination roll-out in this group.

Overall, the results derived from the routine health surveillance network provide valuable insights into the temporal patterns of 21 health related indicators among refugees in one of the largest refugee-receiving countries in Europe. The identified pattern provides evidence of the detrimental impact of the COVID-19 pandemic on health, reflected by a rise in potential consequences of violence (and other external causes) and mental health problems. The only indicator that significantly and consistently declined during the pandemic related to respiratory conditions. We found no sufficiently robust evidence for effects on other indicators.

The strength of our analysis lays in the quasi-experimental situation, in which data was collected before and during the COVID-19 pandemic in a comparable and consistent way by means of a unified EHR. We covered a time period of more than 4 years, and adjusted for potential influences on the outcomes related to individual and centre-related aspects, such as changes in size and composition of denominators. All estimates in the main analysis account for increase or change in the denominator in terms of quantity, but also in terms of composition related to age and sex composition. In descriptive analysis, incidence proportions are weighted based on the denominator (occupancy), while in regression analysis compositional changes are considered by including the proportional change in respective age- and sex-variables.

However, we had no information on provider-related variables, which means that we could not adjust for potential confounding caused by coding behaviour of health professionals, or healthcare routines that may have been affected by the pandemic. Our data was also limited by a lack of information on countries of origin when using occupancy data. To account for compositional changes in countries of origin we hence used the overall number of patients per months and their countries of origin in each of the 21 centres. Despite the use of different data subsets, more than 50% of the estimates (twelve of 21) proved to be robust with respect to the direction of the effect, while the point estimates for the impact of the pandemic (IRR) were higher in all indicators (except pregnancy related conditions) when using occupancy data as denominators (Fig. 2).

As for the risk of compositional effects on our results, the sample is widely representative with respect to age, sex, and distribution of nationalities to the general refugee population in Germany (overview of nationalities see Supplementary File, Chapter 4). An exception are Ukrainian refugees, who are underrepresented in reception centres as a consequence of the freedom of movement granted through EU directives. The average age was about a year lower in the refugee patients after onset of the pandemic, and overall differences broken down by centres (Supplementary File) appear marginal, which makes age differences unlikely as explanation for the changes in incident

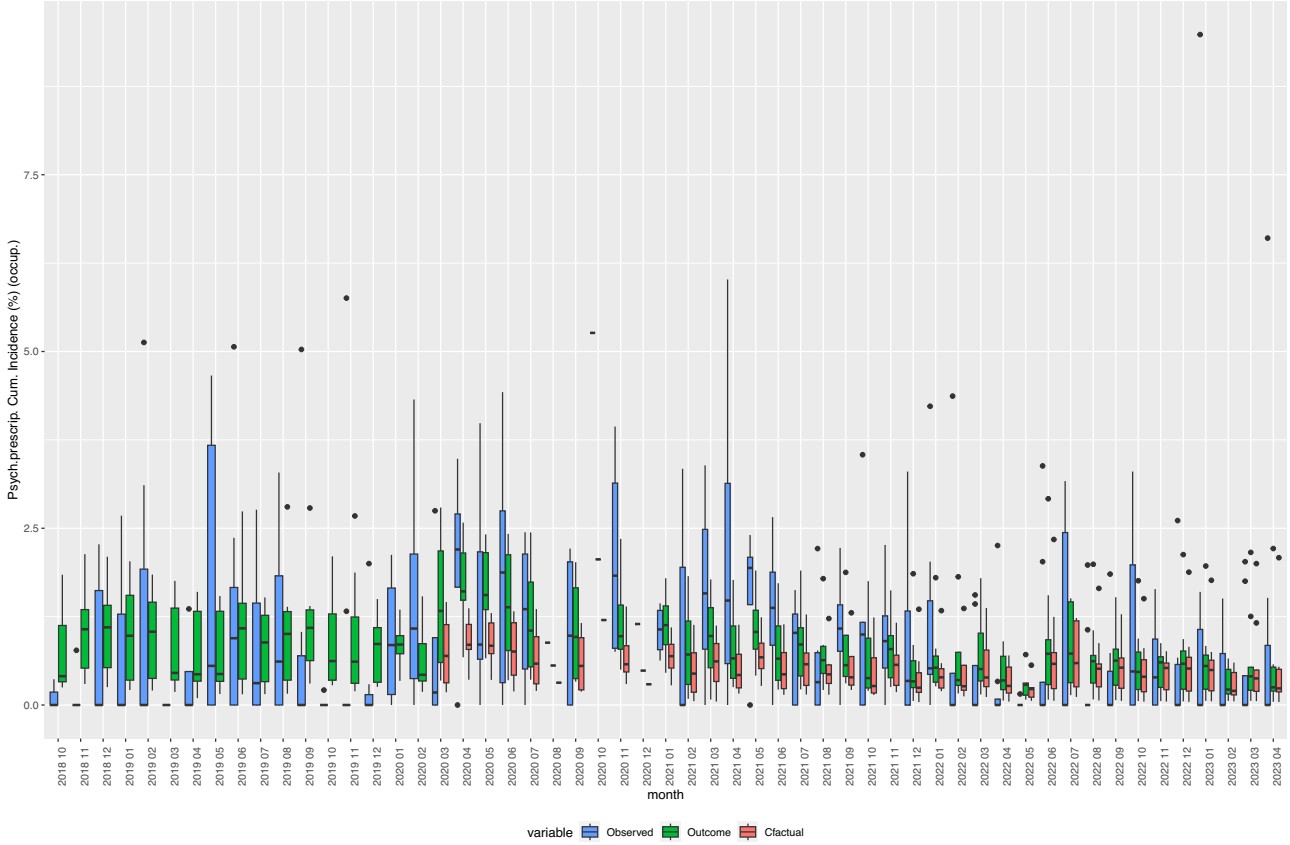

**Fig. 5 | Box Plots of observed, expected, and fitted values of the incidence of "Prescriptions of psychotherapeutic drugs", 2018–2023.** Cfactual: counterfactual (expected) values given the respective age- and sex-distribution of underlying refugee population (occupancy) in reception centres at given time points. Fitted: fitted values based on negative binomial regression models, adjusted for age, sex, centre characteristics, and secular trends. Observed: crude incidence proportions, i.e., cases of individuals with one or more new diagnoses defined by the indicator variable divided by the total number of refugee centre inhabitants (occupancy). Psych. Prescript.: Prescriptions of psychotherapeutic drugs, Cum. Incidence: Cumulative Incidence, occup.: occupancy. Y-axis: cumulative incidence in %. Boxes: interquartile range. Whiskers: Range. Horizontal black bar in boxes: Mediano. N = 836 centre-months. Source data are provided in the "source_data.xlsx" file.

diagnoses. Furthermore, allocation to states, including the three states under study, occurs on a quasi-random manner based on administrative quota, which is why it is very likely that there is no systematic difference refugee characteristics or underlying morbidity between different states. We minimised the risk for residual confounding by adjusting for sex, age, and nationality (amongst others) in our regression models. While reception centres and states may differ with respect to contexts and pandemic measures, we considered this level as random effect in the analysis. While healthcare seeking patterns may have been affected, especially in the beginning of the pandemic, we were not able to account for variation in utilization rates as potential confounder. External validity may also be limited as our sample only covered states in Western Germany. While the risk for compositional bias or residual confounding by compositional factors is low due to above quota and our adjustments, contextual factors in other regions, e.g., political hostility against refugees in Eastern Germany[42] may have led to even higher disadvantages for refugees due to higher levels of racism/hostility towards refugees[42], as well as due to the lower capacity of the regional healthcare system.

Within the federated data analysis design, the study followed-up individuals over time, which is a key strength of the analysis. Each recording of a diagnoses (numerator) is assigned to a single individual in the EHR, and this individual is recognized in any repeat visits and is not double-counted for the calculation of incidence proportions. Incident cases are counted, and incidence proportions are calculated based on unique individual-level data distinguishing cases and the respective population at risk attending the clinics. However, estimates using denominators collected from authorities on a monthly basis are aggregated, so that the population at risk for the calculation of incidence proportions with occupancy data may have been overestimated if individuals remain in the centre for more than a single month, which is very likely. In this case, i.e., in data-subsets 2 to 4 and the main analysis, as well as Sensitivity analyses 2 and 4, our estimates are an underestimation of the true incidence proportion of outcomes, especially in time periods with low in- and out-flows of refugees into the centres, as was the case during the pandemic. Analyses using subset 1 (i.e., Sensitivity analysis 1) based on the electronic health record data alone are not affected by this limitation as the denominator used in these analyses is the number of unique patients per month.

Given the acceptable consistency between findings of the main analysis and sensitivity analyses in terms of direction of effects, we believe the potential underestimation of incidence proportions in occupancy-based analyses does not substantially affect our results regarding the impact of the COVID-19 pandemic on patterns.

Another limitation of our data is the lack of information on length of stay in the centres, which in essence means that we cannot attribute a coded diagnosis to the situation in the centres directly. A condition may be diagnosed in the centre, but be obtained or attributable to experiences made during the migration process, before emigration to Germany or before reaching the centre. However, movement and travel restrictions were very strict in the beginning of the pandemic,

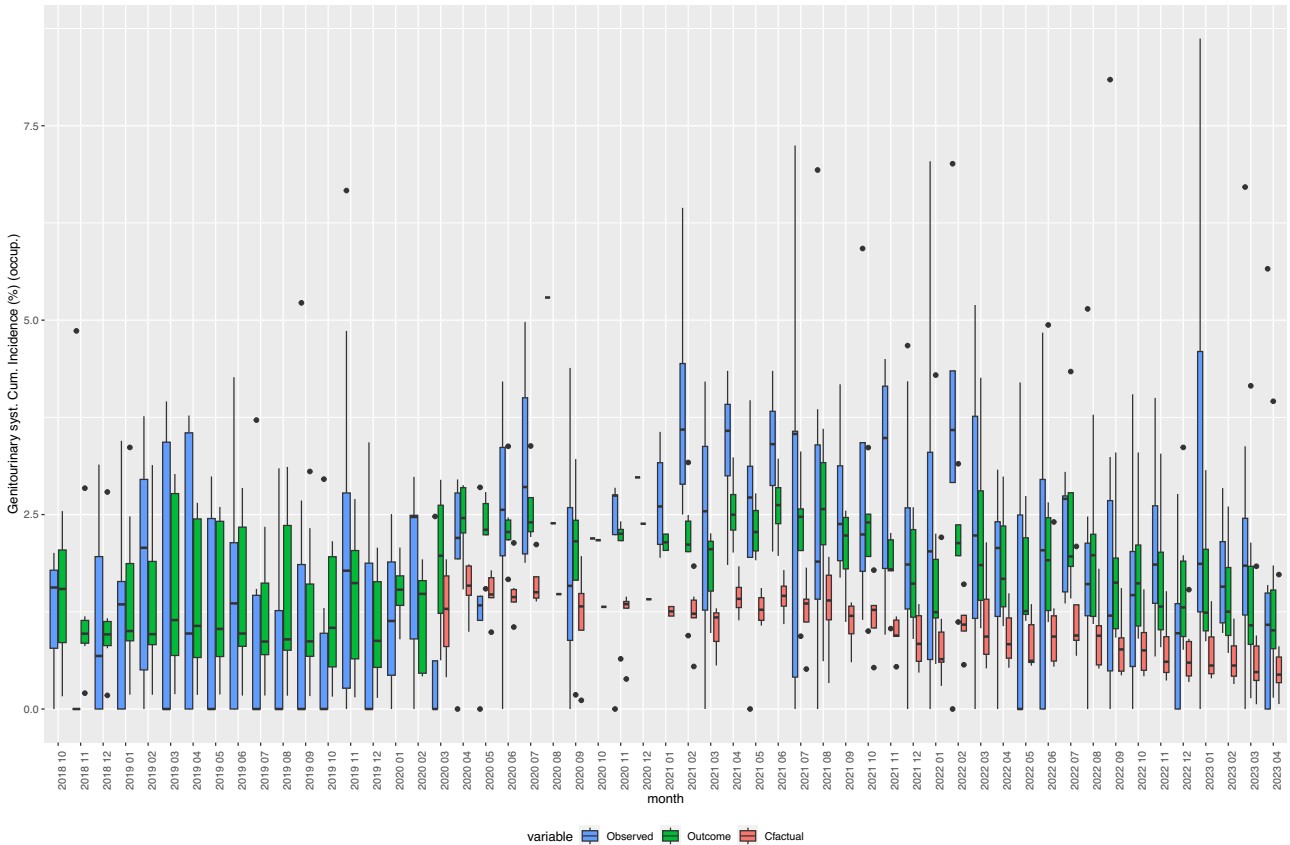

**Fig. 6 | Box Plots of observed, expected, and fitted values of the incidence of "Diseases of the genitourinary system (N00-N99)", 2018–2023.** Cfactual: counterfactual (expected) values given the respective age- and sex-distribution of underlying refugee population (occupancy) in reception centres at given time points. Fitted: fitted values based on negative binomial regression models, adjusted for age, sex, centre characteristics, and secular trends. Observed: crude incidence proportions, i.e., cases of individuals with one or more new diagnoses defined by the indicator variable divided by the total number of refugee centre inhabitants (occupancy). Genitourinary syst. Diseases of the Genitourinary system, Cum. Incidence: Cumulative Incidence, occup.: occupancy. Y-axis: cumulative incidence in %. Boxes: interquartile range. Whiskers: Range. Horizontal black bar in boxes: Median. N = 836 centre-months. Source data are provided in the "source_data.xlsx" file.

and restrictions were only gradually relaxed and lifted, so that the risk of "conflating effects" due to high immigration is unlikely. This is confirmed by national asylum statistics, which shows that asylum applications were comparatively low in 2020, gradually rising to pre-pandemic levels and above until 2023[43].

Another limitation is that we had no data on confirmed COVID-19 cases from all centres. Despite the fact that the EHR contained a module to record COVID-19 testing and test results, only one centre used the module after the onset of the pandemic, while the other centres used parallel systems mandated by respective authorities or public health services to record or notify respective cases. The micro-level fragmentation of health information systems, and the scattered nature of data between immigration and health authorities[17], hence prevented us from considering and analysing the role of COVID-19 outbreaks in changing diagnoses patterns. The divide between health data governance and immigration data governance[17,44] was reflected in our study in terms of different data sources and mandates for health data versus occupancy data in refugee centres. This divide could be partially overcome by linking health data with aggregate data on age, sex, and total numbers of inhabitants of refugee centres, which we prospectively obtained from authorities. However, direct or indirect linkage at individual level would have been beneficial and would have allowed for more robust analyses. This calls for enhanced linkage methods and approaches to overcome the fragmentation of data in the context of migration and health[17].

Further limitations relate to the potential underestimation of incidences of rare events and/or incident diagnoses in small refugees

centres due to our method of anonymisation, which required to set observations less than 3 to zero. While we sought to address this underestimation by applying zero-inflation models (i.e., treating zeros as true values instead of missing data), underestimation cannot be ruled out. However, we accounted for this limitation by weighting our findings based on the size of the occupancy (or patient) population, so that centre-months with larger sample sizes or observations had a higher weight in the estimates. Finally, there were reliability issues with the occupancy data, due to manual entry by authorities and dis-crepancies between totals and age- and sex strata. However, our sensitivity analyses (Supplementary Chapter 2) using different subsets of data show that this did not affect our estimates. The reliability pro-blems in occupancy data underline the relevance of obtaining and capturing reliable denominator data for health studies in the context of forced migration[17,45].

Overall, our study provides robust evidence for changing patterns of morbidity among refugees in refugee centres during the pandemic. Despite localised data sources in three federal states, the insights may be generalisable to other contexts given careful considerations of potential bias and confounding, which are addressed and mitigated in our study design. Patterns changed to the considerable disadvantage of refugees, with higher incident disease burdens related to injury, violence, and mental health conditions. An exception was observed only for respiratory conditions, which declined most likely due to the implemented pandemic measures. Further research is required to better understand the rise in genitourinary conditions. Future pan-demic preparedness and response strategies must better consider the

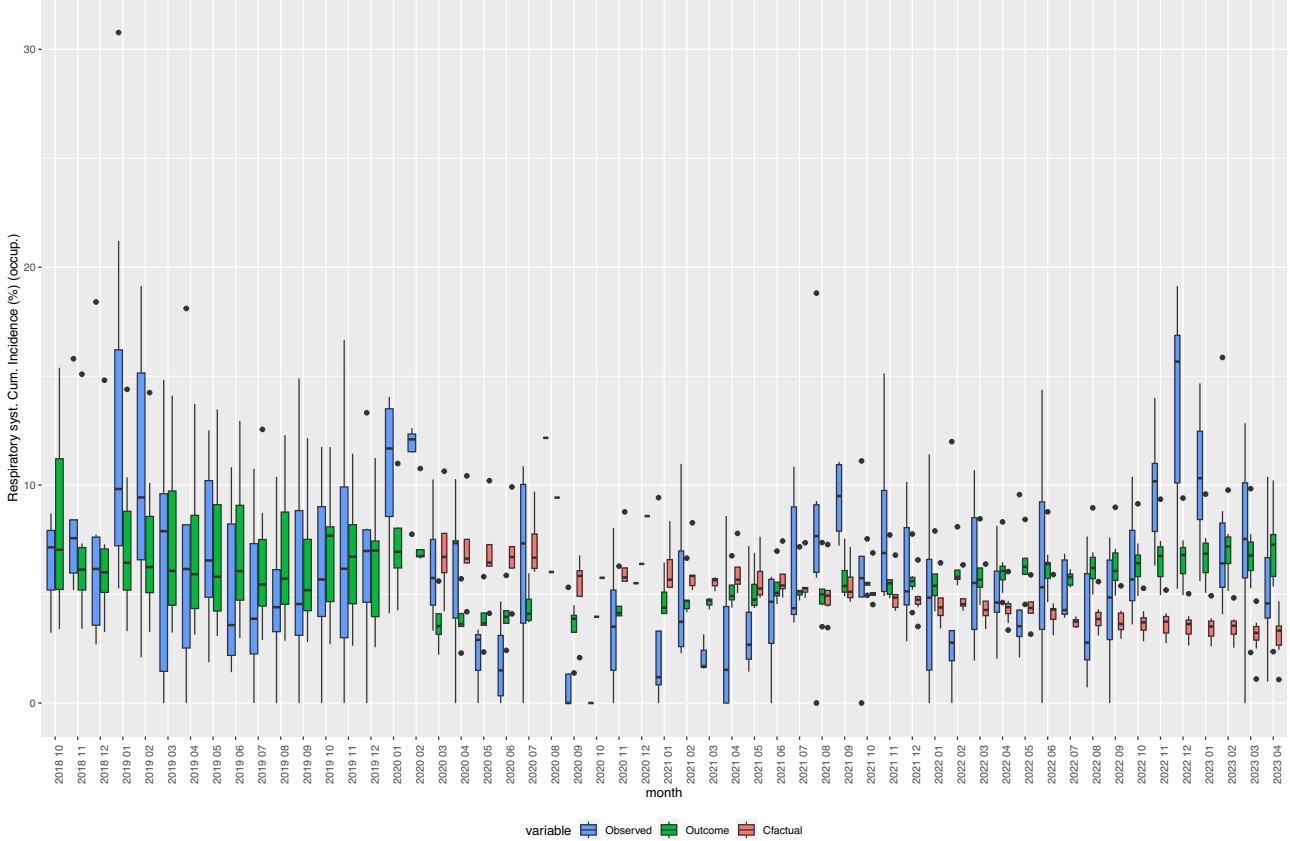

**Fig. 7 | Box Plots of observed, expected, and fitted values of the incidence of "Diseases of the respiratory system(J00-J99)", 2018–2023.** Cfactual: counterfactual (expected) values given the respective age- and sex-distribution of underlying refugee population (occupancy) in reception centres at given time points. Fitted: fitted values based on negative binomial regression models, adjusted for age, sex, centre characteristics, and secular trends. Observed: crude incidence proportions, i.e., cases of individuals with one or more new diagnoses defined by the indicator variable divided by the total number of refugee centre inhabitants (occupancy). Respiratory syst. Diseases of the Respiratory System, Cum. Incidence: Cumulative Incidence, occup.: occupancy Y-axis: cumulative incidence in %. Boxes: interquartile range. Whiskers: Range. Horizontal black bar in boxes: Median. N = 836 centre-months. Source data are provided in the "source_data.xlsx" file.

specific conditions in refugee centres and mitigate the negative consequences of health emergencies.

## Methods

### Study design

The study type is a multi-centre, observational prospective (open) cohort study, analysed in a quasi-experimental (interrupted time series, ITS) design[21]. We use months per centre as observation units, which are obtained from a retrospective (open) cohort of health surveillance data from the Pri.CareNet surveillance network. The impact of COVID-19 on incident diagnosis patterns among refugees was evaluated using a segmented regression approach.

### Setting and data sources

The analysis was conducted within the framework of Pri.CareNet, a health surveillance network[20]. Pri.CareNet is overseen by the University Hospital Heidelberg and comprises healthcare providers operating healthcare facilities on-site as of October 2018. Since December 2023, these facilities are distributed across 24 state-level registration and reception centres, along with one district-level accommodation centre for refugees in Germany. These 25 centres are situated in the German states of Baden-Wuerttemberg, Bavaria, and Hamburg. These states collectively host approximately 30% of the asylum-seeking population in Germany, as determined by administrative quotas[46].

Asylum-seekers are obliged to stay in the reception centres for up-to 18 months or longer before they are transferred to district-level accommodation centres. Asylum-seekers from presumed safe

countries of origin must stay in the centres until the end of their asylum process, but maximum for 24 months. The duration of stay can hence vary considerably depending on stocks and flows, countries of origin, federal state regulations and processes, and the speed of the asylum process. In Baden-Wuerttemberg, state level authorities report an average duration of stay between 8 and 18 weeks, but underlying source data are not public and cannot be validated independently. Other states do not report average durations of stays within centres on a routine basis.

Most reception centres provide on-site health care facilities. These are usually equipped to provide basic primary care services within a heterogeneous infrastructure and with a varying quality, while some centres provide additional services ranging from specialist care to psychosocial services[27,47].

Measures in response to the pandemic comprised decongestion, mass quarantine, as well as reduction of health, social, and other service. Although measures were implemented in a heterogenous way across the country[11,14], the system of centralized reception remained operational, and reception centres were not closed or shut down in favor of decentralized housing[5].

Within Pri.CareNet, healthcare providers are equipped with a customized Electronic Health Record (EHR) system known as Refugee Care Manager (Ref.Care). Ref.Care not only includes standard medical record-keeping features but also incorporates a built-in, federated health surveillance module developed at Heidelberg University Hospital[20,48]. The surveillance module comprises an automated analysis of locally stored medical routine data using predefined indicators.

**Table 2 | Indicator definitions based on diagnoses (ICD-10 Codes) and prescriptions (ATC-Codes) recorded in the electronic health record**

| Indicator labels | Indicator definition | Operationalisation (ICD-10 or ATC-Codes) |
|---|---|---|
| Indicators based on recorded diagnoses | | ICD-10-Codes |
| Disability | Disabilities | H54, R47, H90-H91, H80-H82, Q71-Q73, M20-M21, Z89, G82, F06-F07, I68, P91, F7, F1 |
| Skin | Diseases of the skin and subcutaneous tissue | L00-L99 |
| Cons.ext.causes | Injury, poisoning and certain other consequences of external causes | S00-T98 |
| Digestive syst. | Diseases of the digestive system | K00-K99 |
| Blood | Diseases of the blood and blood-forming organs and certain disorders involving the immune mechanism | D50-D90 |
| Inf.diseases | Certain infectious and parasitic diseases | A00-B99 |
| Inf.notify | Notifiable infectious diseases | B30.0, B30.1, A05.1, A23.0, A23.1, A23.3, A23.8, A23.9, A04.5, A92.0, A00, A81.0, A97, A36, A98.4, A04.4, B67, A04.3, A75.0, A84.1, A95, A07.1, A41.3, A49.2, G00.0, J09, J14, J20.1, P23.6, A98.5, B15, B16, B17.1, B18.2, B19, B16.0, B16.1, B17.0, B17.2, B17.8, B20-B24, D59.3, M31.1, J09, J10, J11, A37, A07.2, A96.2, A68.0, A48.1, A48.2, A30, A27, A32, P37.2, B50-B54, A98.3, B05, A39, A41.0, A49.0, G00.3, P36.2, A22, B26.8, B26.9, A08.1, A70, A01.1, A01.2, A01.3, A01.4, A20, A80, A78, A08.0, P35.0, B06.8, B06.9, A0, A03, A50, A53, A82, Z20.3, P37.1, B75, A15 - A19, P37.0, O98.0, A21, A01.0, A92.0, A92.4, A96, A98.0, A98.1, A99, B02, P35.8, A04.6 |
| Circulatory syst. | Diseases of the circulatory system | I00 – I99 |
| Hypertension | Hypertension | I10-I15 |
| Metabolic | Endocrine, nutritional and metabolic diseases | E00-E90 |
| Diabetes | Diabetes mellitus | E10-E14 |
| Musculoskelet. syst. | Diseases of the musculoskeletal system and connective tissue | M00-M99 |
| Neoplasm | Neoplasms | C00-D48 |
| Nervous syst. | Diseases of the nervous system | G00-G99 |
| Ear.mastoid | Diseases of the ear and mastoid process | H60-H99 |
| Eye.adnexa | Diseases of the eye and adnexa | H00-H59 |
| Pregn.condition | Pregnancy, childbirth and the puerperium | O00-O99 |
| Psych.condition | Mental and behavioral disorders | F00-F99 |
| Genitourinary syst. | Diseases of the genitourinary system | N00-N99 |
| Respiratory syst. | Diseases of the respiratory system | J00-J99 |
| Indicators based on recorded prescriptions | | ATC-Codes |
| Psych. prescrip. | Psychoactive drug prescriptions | N05, N06A, N06B, N06C, N07BB |

*ICD-10* International Classification of Disease, *ATC* Anatomical Therapeutic Chemical Classification.

The indicators are constructed using diagnosis categories based on International Classification of Diseases (ICD-10-GM Version 2021) and drug prescriptions based on the Anatomic Therapeutic Classification (ATC 2023) as defined and outlined in Table 2, and operationalized through a standardised analysis script[20,48]. Indicators were developed and consented within the consortium by practitioners, healthcare providers, and authorities in a collaborative research approach. To protect data anonymity, any observations with counts less than 3 are adjusted to 0. More detailed information about the surveillance infrastructure in Pri.Care*Net*, and the local analysis of indicators can be found in previous reports[20,48].

The data used in this paper was generated using Ref.Care version 1.1.8 (ref.care) and covers the time span from October 2018 to April 2023. The facilities included in this study joined the surveillance network at different dates (Supplementary Chapter 3). Some centres have since departed from the network due to closures or changes in healthcare providers, but still contributed their anonymous health surveillance data for the purpose of this study. Provided data consequently varies per centre (Supplementary Chapter 3).

Ref.Care is used by health professionals, who are the data holders of the individual-level patient data in on-site health care facilities. They enter the medical data into the EHR as part of their routine clinical work, so that no parallel data entry effort is required. Data entry fields are both standardized and allow for free texts (e.g., for medical history), but all data used in this analysis is based on standardized codes and diagnoses (see below). The respective authorities in the three federal states are responsible for immigration data, and are data holders of the occupancy data, i.e., the sociodemographic information of the refugee centres' inhabitants.

The flowchart in Fig. 8 provides an overview of the data selection process, the nature of used data sources and the four derived data subsets (Subset 1–4).

Individuals are included in the medical records if they seek care in one of the on-site medical clinicals established for refugees, so that data are "utilisation data", which means that health data of individuals who did not seek care is not included. Individuals may also seek care outside of reception centres, but this usually requires approval, and is linked to several geographic, linguistic and administrative barriers, which is why on-site services usually constitute a low-threshold and first anchor point for patients seeking care. In case of serious and chronic conditions, data of referrals or re-referrals to providers of the regular health system are recorded in Ref.Care in the refugee centres as well to ensure ongoing treatment, so that medical history can regarded as fairly complete. However, it cannot be ruled out that major diagnoses have been not recorded in case of treatment outside of the facilities in the regular health system.

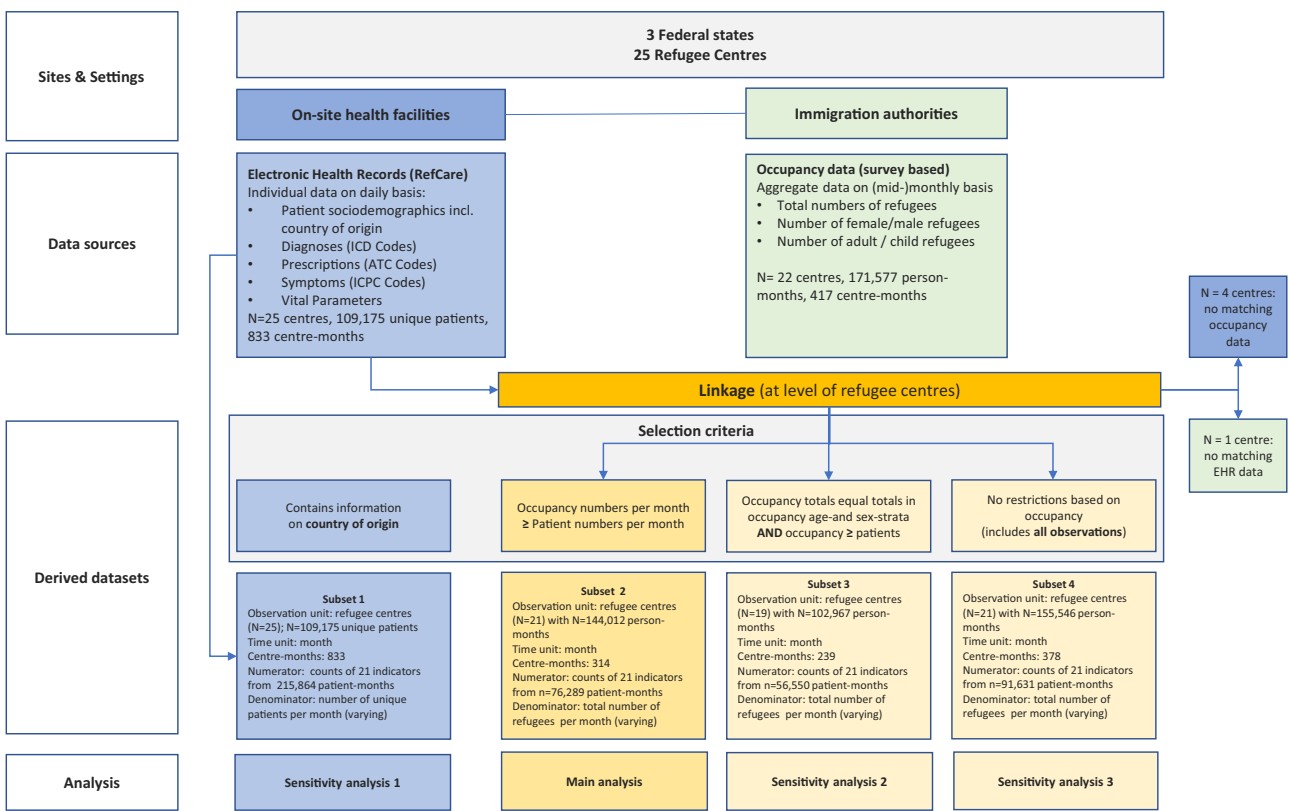

**Fig. 8 | Flow Chart with overview of sites and settings, data sources, and derived datasets for analyses.** Ref.Care Refugee Care Manager, ICD-10 International Classification of Disease, ATC Anatomical Therapeutic Chemical Classification, ICPC International Classification of Primary Care, EHR electronic health records.

## Electronic health records (Ref.Care) data set (subset 1)

Using the 25 refugee centres and months as units of analysis, subset 1 contains 833 observations (i.e., 833 "centre-months") of recorded medical data with an average of $mean(n_{pat}) = 259$ (1) (standard deviation $sd(n_{pat}) = 287$) (2) patient-months. The sample comprised 215.864 patient-months ($= \sum_{i=1}^{833} n_{pat}^i$, (3) where $n_{pat}^i$ is the number of refugee patients of "centre-month" $i$) of a total of 109.175 refugee patients between October 2018 and April 2023 (Fig. 8). For these 833 centre-months, we used reported monitoring data on the number of male, female, adult (≥18 years of age) and underage (<18 years of age) patients; as well as data on the incident coding of diagnoses for 21 indicators (based on ICD-10 Codes) by centre and month. For each centre, age data is available in 5-years age groups. To calculate the mean age of the patient population before and after onset of the pandemic, we divided data into time periods before and after onset of the COVID-19 pandemic. Then, for each facility and overall, we calculated the frequency of each age group for the periods before and after March 2020, and multiplied these values by the mean value of each age group category. We then summed up these values and divided them by the number of patients in each facility or all patients combined to calculate the average age before and after March 2020 (Supplementary Chapter 3.3). As differences appeared marginal (difference in means: 1.0 year, standard deviation 1.7), we used age as dichotomous variable (proportion of adults: ≥18 years and proportion of children: ≤18 years) in the regression models.

Furthermore, in Sensitivity analysis 1, we used data on the country of origin of the patients from the EHR to run models which account for compositional differences in the refugee population within and between refugee centres over time (Supplementary Chapter 2.1). Estimates for the COVID-19 impact from this sensitivity analysis are reported in Fig. 2.

## Occupancy data and aggregate-level socio-demographics

Furthermore, we gathered information on occupancy of each refugee centre within the Pri.CareNet surveillance network through a monthly online survey conducted with the responsible authorities of these centres. This prospective census survey was initiated in October 2018 and encompasses count data concerning the number of residents on the 15th day of each respective month, categorized by age (adults: ≥18 years and children: ≤18 years), and biological sex (male/female) (Fig. 8). To determine the total occupancy of each centre for every month, we combined the reported counts of male and female adults separately for the adult population and likewise for the children. These cumulative counts of children and adults were then summed to calculate the total occupancy for each centre and month. Furthermore, the overall (unstratified) number of the occupancy was collected.

Participation of authorities in this survey is voluntary. We collected occupancy data from 22 centres, resulting in a comprehensive dataset covering 417 centre-months spanning from October 2018 through June 2023 (Fig. 8). The average occupancy stands at $mean(n_{occ}) = 411$ (4) individuals per centre per month, with a standard deviation of $sd(n_{occ}) = 435$ (5).

## Description of derived datasets and variables

We matched the EHR data with the monthly occupancy data for each centre, wherever possible (Fig. 8). In 64 cases, the occupancy count was lower than the number of patients (i.e., $n_{occ} < n_{pat}$) (6). This occurrence is reasonable in situations where refugee centres experience a rapid turnover of individuals, such as a high influx of new arrivals and frequent transfers. In such instances, individuals may seek on-site healthcare services but stay within the centres for only a brief period, leading to a temporary misalignment between occupancy figures and the number of patients receiving healthcare services. These observations were excluded for the main analysis which resulted in a total of 314 centre-months between October 2018 and April 2023 of 21

centres (with $mean(n_{pat}) = 243$ (7), $sd(n_{pat}) = 240$ (8), $mean(n_{occ}) = 459$ (9) and $sd(n_{occ}) = 462$ (10); subset 2).

In 75 cases, the sum of the reported strata counts (female/male x adult/children) did not equal the reported total occupancy. Therefore, we repeated the main analysis on subset 3 (Sensitivity analysis 2), where the occupancy totals equal the totals in occupancy age-and sex-strata AND $n_{occ} \geq n_{pat}$ (11) (Supplementary Chapter 2.2). Furthermore, in Sensitivity analysis 3, we repeated the main analysis again (which was performed on subset 2), but instead used subset 4 of the linked data, which contained no restrictions, i.e., all observations of the linked dataset (Supplementary Chapter 2.3).

Furthermore, we calculated the following variables (for each subset, respectively):
- time: discrete variable indicating time from the start up to the end of the observation period October 2018 to April 2023 with time ID = {1, ..., 56} (12)
- covid: coded 0 for pre-covid time points and 1 for post-covid time points (0: < March 2020, 1: ≥ March 2020). This variable captures the impact of the COVID-19 pandemic in peri-pandemic time periods, with pre-pandemic time periods used as reference.
- postslope: coded 0 up to the last point before COVID-19 and coded sequentially from 1 thereafter (0: < March 2020, 1: March 2020, 2: April 2020, ..., 37: April 2023). This variable captures the peri-pandemic time trend.

It should be noted that there are two levels in the data: months and centres. As a result, there are multiple observations of these levels per year. Therefore, in order to report the mean incidence of analysed outcomes (Table 1), we determined weighted mean values averaging over the months for each facility per year, so that there is only one observation per year of a facility. The weighting was based on $n_{occ}$, i.e., months with high occupancy were assigned a higher weighting when calculating the weighted mean incidence for each facility ("weighted mean facility observation"). Table 1 shows the mean value with standard deviation (mean ± sd), median with 25th and 75th quartiles (Q1, Q3), minimum and maximum (min–max) and 95% confidence interval (CI) weighted mean values of facility observations. Furthermore, the annual weighted mean value and weighted standard deviation are given, whereby the weighting was accordingly to the mean occupancy of a facility within one year. That is, if the mean occupancy of a facility in one year is higher, the observation weighs more ("weighted annual": mean and standard deviation of the "mean facility observation" values within one year weighted by the mean occupancy of the respective facility; compare row W, 2018–2023). Additionally, the mean value and the standard deviation of the weighted annual mean values were calculated (row W, last column). Indicators with very few observations per centre-month (e.g., congenital disorders, multimorbidity) were excluded from the analysis to avoid problems with convergence and fitting of regression models.

We implemented additional sensitivity analyses with the attempt to capture and evaluate nuances during the peri-pandemic phase beyond a dichotomous approach described above.

In Sensitivity analysis 4, we implemented a more fine-grained analysis that distinguishes three phases of the pandemic:
-Pre-pandemic phase (before 02/2020)
-Phase 1 of the pandemic: representing the early phase until the onset of vaccine availability, i.e., 02/2020–07/2021
-Phase 2 of the pandemic: Emerging variants phase (e.g., delta, omicron, etc.) and increasing vaccine role out, i.e., 08/2021 to 04/2023
The results are presented in Supplementary File, Chapter 5 (Fig. S26).

In Sensitivity analysis 5, we have implemented an analysis to examine seasonality effects, comparing effects of spring, autumn, and winter periods with summer periods (2018–2023), respectively. The results are presented in Supplementary File, Chapter 5 (Fig. S27),

showing that seasonality effects exist in particular for consequences of external causes of injuries, with higher incidences in summer periods compared to spring, winter and autumn, respectively. This may be due to route dependent effects (more dangerous routes during summer), heat related effects on violence, or higher probability of injuries related to outdoor activities. The incidence of psychological conditions was higher in winter periods (compared to summer periods), while respiratory conditions were significantly higher in winter and autumn (compared to summer periods). No considerable seasonality effects were observed for the remainder of diagnoses. The patterns prove the high plausibility of the surveillance data, and are overall consistent with the patterns observed in the main analysis.

### Description of the regression model
In order to assess the impact of the COVID-19 pandemic on the incident health indicators, we fitted a negative binominal model with zero-inflation model on the matched data for each indicator. The model allows the conditional mean to depend on the percentage of adult and male occupancy, overall number of occupancy ($n_{occ}$) as well as randomly on centres, while $\beta_0$ captures the baseline level of the outcome at time 0 (beginning of the observation period), $\beta_{time}$ estimates the structural trend or growth rate, independently from COVID-19, $\beta_{covid}$ estimates the immediate impact of COVID-19 or the change in the outcome of interest after COVID-19 and $\beta_{postslope}$ reflects the change in the trend or growth rate in the outcome after COVID-19. Furthermore, the model assumes structural zeros (Supplementary Chapter 1). The model can be represented by the following set of equations:

$$\mu = E(count, |, u, NSZ) = \exp\left(\beta_0 + \beta_{adult} + \beta_{male} + \beta_{n_{occ}} + \beta_{time} + \beta_{covid} + \beta_{postslope} + u\right),$$
(13)

$$u \sim \mathcal{N}(0, \sigma_u^2),$$
(14)

$$\sigma^2 = Var(count | u, NSZ) = \mu\left(1 + \frac{\mu}{\theta}\right),$$
(15)

$$logit(p) = \beta_0^{(zi)}$$
(16)

where $u$ is a centre specific random effect, NSZ is the event "non-structural zero", $p = 1 - Pr(NSZ)$ (17) is the zero-inflation probability and $\beta$'s are the regression coefficients with subscript denoting the covariate and with 0 denoting the intercept[49]. The chosen parameterization of the negative binomial uses a logarithmic link and denotes the variance increasing quadratically with the mean as $\sigma^2 = \mu(1 + \mu/\theta)$ (18), with $\theta > 0$ [50] (19) (Supplementary Chapter 1).

### Counterfactual analysis
We performed a counterfactual analysis by predicting the expected values of the 21 health indicators given that the pandemic had not happened (variable covid set at "0") while considering the socio-demographic characteristics of the underlying refugee population in respective centres and time periods. We plotted the estimated counterfactual, observed, and expected incidence rates in percent (i.e., the number of cases divided by occupancy and multiplied by 100) of selected indicators together in box plots over the observation period (compare Figs. 3–7).

### Sex and gender considerations
Sex is based on data coded in routine medical records (based on physicians' coding), gender was not captured or available. Sex was considered in the analyses as co-variable in regression models. Differential effects of the pandemic between men and women were not

analysed. Sex-stratified descriptive data of all outcomes are available in the Supplementary Material.

## Race, ethnicity and other social categories of the study population

No variables of race or ethnicity were used. Country of origin, i.e., nationality recorded in the medical records system, was used in the regression models to adjust for differences in incident diagnoses that may be attributable to compositional changes of the underlying refugee population in a centre, rather than to the analysed exposure (COVID-19 pandemic). Nationalities included in the analyses were restricted to those countries with the most frequent share among the refugee population across all centres (Afghanistan, Iraq, Nigeria, Syria, Turkiye). We use the social category *refugee* in our manuscript and the analysis, acknowledging that this category subsumes a heterogenous population. We further use this term as umbrella term for persons who are registered in the centres and live there with different residence status as asylum claimants, asylum seekers, resettlement refugees, accepted refugees, or tolerated individuals. As our data is based on medical records data, no information was available for the different residence status to allow for more nuanced disaggregation, or in order to evaluate and assess any within-group variations among the social category of *refugees*. The study population did not contain general migrants, but was confined to refugees living in camp-like or institutionalised accommodation facilities (reception centres).

## Statistical software

All analyses were conducted in the R-programming language version 4.2.1 using the glmmTMB-package[50] for fitting mixed-effects models.

## Ethics

The study uses de-individualised anonymous (aggregate) data, generated from a federated data analysis methodology performed on individual-level clinical data with the result of anonymous counts. The methodology and implementation of the data protection approach for federated data analysis has been reviewed and approved by the Review Board of the Technology and Methods Platform for Networked Medical Research (TMF e.V.). The study has been additionally reviewed and approved by the Ethics Board of the Medical Faculty of Heidelberg University (S-646/2024).

## Reporting summary

Further information on research design is available in the Nature Portfolio Reporting Summary linked to this article.

## Data availability

The datasets generated and/or analysed during the current study are not publicly available due to the data-use and -access (DUAC) regulations of the Pri.Care*Net* Consortium. The generated and analysed datasets are available for scientific purposes from the Pri.Care*Net* Consortium upon request by contacting the spokesperson (Kayvan Bozorgmehr, refcare.AMED@med.uni-heidelberg.de). Data provision is subject to written request with a specification of detailed research questions, a draft analysis plan, the cooperation with at least one consortium member, as well as clearance by the Consortium's DUAC based on criteria of feasibility and ethical considerations. Source data are provided with this paper.

## Code availability

The original R output can be found in the Supplementary File. The full code underlying the analyses in this report can be obtained upon request (Kayvan Bozorgmehr, refcare.AMED@med.uni-heidelberg.de).

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

## Acknowledgements

We would like to thank all participating reception centres for asylum seekers for supporting the establishment of the Pri.Care Surveillance Network (Pri.Care*Net*-Consortium).

## Author contributions

K.B. conceived the study. K.B. and S.E. designed the statistical methodology. S.E., S.R., K.B., and R.J. collected and curated the data. S.E. prepared the data, conducted the analysis and created figures and tables. K.B. and S.E. wrote the first draft of the manuscript, K.B. wrote

revised versions and the final draft. S.E., S.R., and R.J. reviewed and edited revisions for important intellectual content. K.B., R.J., and S.R. verified data analysis and validated the findings. The Pri.Care*Net*-Consortium has made important contributions to data curation and implementation (V.W., A.W.G., J.K., L.C.), and/or critically reviewed the manuscript for important content (A.H., O.R., S.R.). All authors have access to the data in the study and had final responsibility for the decision to submit for publication.

## Funding

## Competing interests

Financial competing interests: The authors acknowledge research support and institutional funding received by the German Federal Ministry of Health in line with a resolution passed by the German Bundestag (Grant no: 2516FSB415, Grant holder: K.B.) in the period 2016–2020. Funds were received for salaries and equipment to develop, validate and implement the surveillance methodology, technology, and infrastructure. We acknowledge further funding received by the State Ministry of Justice and Migration (Baden-Württemberg) and Regional Authorities in Bavaria as well as care provider organisations (Klinikum Würzburg, St. Joseph Klinik, MKT) for operational running costs of the use and implementation of the electronic medical records software Ref.Care in the scope of the surveillance network within a non-for-profit licensing model (Grant holder: K.B.). The funders had no role in design, analysis, or interpretation of data or in the decision to publish. Personal financial interests: K.B. and R.J. are registered at University Hospital Heidelberg as co-inventors of the electronic medical records software Ref.Care in line with the Employee Invention Act (ArbnErfG). The invention is related to the underlying software and concept for surveillance, without receiving any individual financial benefits from licenses or use and implementation of the software. The authors declare no other competing interests.

## Additional information

## Consortium Pri.CareNet

**Veronika Wiemker[2], Andreas W. Gold[2], Lena Conz[2], John Krueger[2], Anna Hansel[4], Oliver Razum[5] & Siegbert Rieg[6]**

[4]Division of Infectious Diseases, Outpatient Refugee Clinic, University Medical Center Freiburg, Freiburg, Germany. [5]Department of Epidemiology & International Public Health, School of Public Health, Bielefeld University, Bielefeld, Germany. [6]Division of Infectious Diseases, Clinic for Gastroenterology, Hepatology, Endocrinology and Infectious diseases & IFB-Center for Chronic Immunodeficiency (CCI), University Medical Center Freiburg, Freiburg, Germany.

