## [Transparent Peer Review file · Nature Communications]

Impact of the COVID-19 pandemic on incident diagnoses in German refugee centres 2018 to 2023

Corresponding Author: Professor Kayvan Bozorgmehr

Version 0:

Reviewer comments:

Reviewer #1

(Remarks to the Author)

Thanks for sending me this paper, which used records from the Refugee Centers in Germany to use the pandemic as a natural time series experiment for assessing the incidence of new diagnoses among refugees during 2020-2023. It shows that during the pandemic, there were a higher number of diagnoses of injury, violence, and mental health conditions than before the pandemic.

Originality: This is an original paper that highlights new and noteworthy results about refugee health during the pandemic. Refugee populations are difficult to reach, and this database is unique and allows a comprehensive understanding of incident conditions.

Methods:

(1) How did the analysis account for the increase or change in the denominator of the population?

(2) Were they the same individuals followed up over time, or was it a mixture or was the study unable to tell individuals from another? The text within the main manuscript is not clear in this regard. If it was "hospital count" data then make this clearer.

(3) The methods could use more detail on these centers and setting – how long are stays usually in the center? Where do refugees go after the center? What happened to the other 70% of centres in Germany during the pandemic - did these remain open or closed, which could have impacted centre population demographics?

(4) How were indicators chosen and prioritised?

Although it is unlikely, a change in the demographic or more refugee admissions could be an alternative explanation for the increase in certain conditions, this remains a bias in the study design.

Parts of the paper are quite difficult to read and follow. Whilst the details provided in the Nature Communications summary on the study design were very clear - I think these pieces of text need to be integrated into the methods section (both in the manuscript and the supplementary section).

(Remarks on code availability)

Reviewer #2

(Remarks to the Author)

What are the noteworthy results?

During the pandemic, among refugees there was an increased incidence of injury, violence and mental health conditions. Declined incidence was seen in respiratory conditions.

Will the work be of significance to the field and related fields?

It will be of limited significance, as most of the findings include some uncertainty, and the findings are mainly in line with what has been already known.

How does it compare to the established literature?

The results are in line with the established literature.

Does the work support the conclusions and claims, or is additional evidence needed?

The work supports the conclusions and claims.

Are there any flaws in the data analysis, interpretation and conclusions? Do these prohibit publication or require revision?

I did not find any flaws in the data analysis, interpretation or conclusions.

Is the methodology sound?

I find the methodology used sound for this purpose.

Does the work meet the expected standards in your field?

The work meets the expected standards, the results are presented in a clear way and the language is excellent.

Is there enough detail provided in the methods for the work to be reproduced?

Enough details for the methods are presented.

I do not have any specific suggestions for changes or any errors to be corrected.

As a general comment, I would stress that using diagnosis codes as the indicator for changes in disease patterns over time during an pandemic has several uncertainties:

- The care seeking patterns were probably influenced.
- The access to care was probably influenced.
- The health care routines and personnel at the centers was probably affected. This might have changed the coding behaviours and the diagnostic criteria.
- No comparison group was used, only temporal changes were analysed. One interesting approach would have been to compare changes in incidence between refugees with the general population.

I suggest that these limitations would be better reflected in the discussion, and if possible some data presented about these factors (number of visits, changes in personnel, access to care, waiting times etc).

(Remarks on code availability)

Reviewer #3

(Remarks to the Author)

I am writing to comment on the research article "The Impact of the COVID-19 Pandemic on Morbidity Patterns in Refugee Accommodation Centers in Germany" which was recently submitted to your journal. While the study provides valuable insights into the health impacts of the pandemic on refugees using a unique natural experiment design, I found the paper extremely difficult to follow due to several issues.

The authors made excessive use of technical terminology without clear definitions. Terms like "zero-inflated negative binomial models", "peri-pandemic", "quasi-experimental situation", and "interrupted time series design" were used without explaining what they mean for non-expert readers. This made it challenging to understand the methodology and analysis. The authors found an increase in incidents of injuries, mental disorders, 39 psychotherapeutic drug prescriptions, and genitourinary diseases and a decrease in respiratory diseases.

More demographic details about the refugee population would have helped interpret the findings. Beyond just gender and adult status, providing information on countries of origin may explain why injuries and mental health problems rose so much during the pandemic period. It is also unclear how refugees were permitted to enter Germany during this strict time.

The authors could have discussed whether the refugee population tended to be older prior to COVID-19 and if age could

explain some of the health impacts found.

It was also surprising to see a decline in respiratory disease reported without further explanation. Including data on the refugee's country of origin and COVID-19 infections rate themselves would have been insightful.

The graphs presented were difficult to interpret without labels on the categories and clearer axis labels.

Comparing the findings to general population health trends in Germany and globally could provide important context.

Overall, while the study design was valuable, clearer presentation of methods, definitions, data set details, population details and discussion of implications are needed for readers to fully understand and evaluate results. I had to read it several times to understand and the end of the day, it's hard to judge how the findings for refugees in Germany compare with the general population.

(Remarks on code availability)

Reviewer #4

(Remarks to the Author)

The manuscript entitled "Impact of COVID-19 pandemic on incident diagnosis patterns in German refugee centres: quasi-experimental study, 2018-2023" presents unique surveillance data to compare morbidity patterns among refugees in Germany before, during and after the COVID-19 pandemic, covering the timeframe from October 2018 to April 2023. The paper is generally well written and the analyses are rigorously conducted. I see high scientific value in the paper given the scarcity of evidence we have so far on refugee and migrant health. I would generally consider the paper as suitable for Nature Communications. However, I advise the authors to add further clarity and context to the paper, following the points I outline below.

Major comments:

1. There is some conflation between the terms migrants and refugees. The authors need to specify what population they look at. If I understand correctly, their sample does not include migrants who have resided in Germany for several years but only refugees who are waiting for their asylum cases to be approved? But this is not clearly spelled out and should be explained in more detail.

2. The authors should provide more context on how refugees in Germany are usually integrated into the formal health care system. Is all medical care they receive delivered on-site or is it common for refugees to also attend regular primary care facilities (Hausarztpraxis)? Related to this, it needs to be specified whether we can expect that the PriCareNet likely records the complete medical history of the refugees upon their arrival in Germany or whether some refugees may have used other care services/facilities and would thus not appear in the PeriCareNet records.

3. While the authors provide detailed information on how COVID policies related to testing, quarantining and mask wearing affected refugees, there is no mentioning of how the German vaccination campaign considered refugees. Were these groups prioritised in the administration of vaccine doses or did they only get later access to vaccination?

4. The dataset only includes information from refugee centers in three German states (Bundesländer). The authors need to more explicitly acknowledge this upfront (i.e. in the abstract or introduction), given that the presented data can therefore not be considered as representative for Germany as a whole. Relatedly, the authors should also extend their limitation section by a discussion of the external validity of their results (to the other 13 German states), especially considering that the data they present is from West Germany and that refugees in East Germany may even face higher disadvantages due to both lower capacity of the healthcare system in Eastern German states as well as higher levels of racism/hostility towards refugees.

5. The analysis includes aggregated information on characteristics of the included refugee centers, including occupant numbers, composition of adult and child refugees, and composition of male and female refugees. I believe that it would add great value if the authors tried to add additional information on the included refugee centers, including (1) proxies for overcrowding/inadequate shelter (i.e., squaremeter per occupant or number of rooms per occupants?), (2) number of staff working in the center/ratio staff:occupants, and possibly (3) annual per capita budget/funding available to the refugee center.

6. The authors should elaborate more in detail how the data is entered into the PeriCareNet database. Who enters the data? How often is data entered? How accurate and complete is the entered data?

7. On p. 8, the authors first point to the lack of psychosocial care services in most reception centers (line 228) but then observe an increase in prescriptions of psychotherapeutic drugs (line 231). There needs to be a more detailed explanation of which psychiatric and counselling services refugees in Germany usually have access to (and how/which kind of these services may have been constrained/disrupted due to the pandemic).

8. The authors observe an increase in genital and other infections during the pandemic and largely interpret this as a consequence of COVID infections. However, I was wondering whether the pandemic may also have caused changes in sexual behaviours, greater exposure to sexual violence, and disrupted access to sexual health care services, which may

serve as an alternative explanation for the observed increase. See for example:

<https://link.springer.com/article/10.1186/s12889-022-13866-7>

<https://www.tandfonline.com/doi/full/10.1080/26410397.2020.1822493>

<https://gh.bmj.com/content/7/10/e009594.abstract>

9. The authors treat the peri-pandemic phase as fully homogeneous as their analysis only differentiates - following a binary logic - between pre-pandemic (0) and post-pandemic months. However, the timeframe from the onset of the COVID-19 pandemic until April 2023 was very heterogeneous in terms of COVID-19 infection numbers, mortality rates and - linked to this - policy measures such as lockdowns or contact restrictions that were adopted to contain the spread of the virus. Would it make sense to include a more fine-grained analysis (at least as sensitivity checks) that tries to model the "strict" and "loose" phases of the pandemic (e.g. drawing on Google mobility data) in a more nuanced way? This could also help establish whether increases in certain health outcomes (e.g. mental health diagnoses) are more substantial during/after times with tougher restrictions.

Minor comments:

1. Introduction: The introduction is well written and nicely motivates from a health equity perspective why it is important to focus on migrant and refugee health and why refugees might be particularly vulnerable to facing adverse consequences due to the pandemic. I suggest that the authors extend the second paragraph (p.3) to also note that migrant/refugee populations may face health disadvantages during pandemics due to their disintegration into the regular health care system and language barriers/health literacy barriers that may hamper their access to necessary health information.
2. Introduction, paragraph 3: "... and quarantine upon transfer" - the authors should specify what kind of transfer they refer to here. Is this the transfer from one refugee center to another?
3. Discussion, p. 8, line 220 - the authors attribute the rise in injuries partly to experiences during flight, however, this would be a factor independent of the COVID-19 pandemic and would have occurred before the patients joined the refugee center. Why would violence during the flight have increased in consequence of the pandemic? This is unclear.
4. Discussion, p. 8, line 225: "... and other adverse experiences" - please specify these.
5. The authors should possibly give some examples of what "diseases of the genitourinary system" could be since this is likely not common knowledge among most readers.

(Remarks on code availability)

Version 1:

Reviewer comments:

Reviewer #1

(Remarks to the Author)

Thanks for addressing my comments. I have no additional comments.

(Remarks on code availability)

Reviewer #4

(Remarks to the Author)

I thank the authors for the very thorough revision of their manuscript and the additional sensitivity analyses conducted, which all confirm the robustness of the results. All my comments have been adequately addressed and the responses have been well explained. I just have one very minor remaining comment. The authors write: "Nationalities included in the analyses were restricted to those countries with the most frequent share among the refugee population across all centres." - could the authors list these nationalities? I believe it's likely Iraq, Afghanistan, Syria, Ukraine, Somalia...? What else? Other than this, I really like the paper and am looking forward to seeing it published soon.

(Remarks on code availability)

POINT BY POINT RESPONSE TO REVIEWERS

Reviewer #1 (Remarks to the Author):

Thanks for sending me this paper, which used records from the Refugee Centers in Germany to use the pandemic as a natural time series experiment for assessing the incidence of new diagnoses among refugees during 2020-2023. It shows that during the pandemic, there were a higher number of diagnoses of injury, violence, and mental health conditions than before the pandemic.

Originality: This is an original paper that highlights new and noteworthy results about refugee health during the pandemic. Refugee populations are difficult to reach, and this database is unique and allows a comprehensive understanding of incident conditions.

We thank the reviewer for the comment and judgement on the relevance of the findings and the data.

Methods:

(1) How did the analysis account for the increase or change in the denominator of the population?

We accounted for changes in the denominator of the population (= occupancy in refugee centres) by linking the medical data from electronic health records (EHR) to the occupancy data prospectively collected via surveys from authorities (see methods page 22, "Occupancy data and aggregate-level socio-demographics"). This matching exercise yielded subset 2 of our dataset, which was used for our main analysis and for the calculation of incidence proportions. In other words: all estimates in the main analysis account for increase or change in the denominator in terms of quantity, but also in terms of composition related to age and sex composition. In descriptive analysis, incidence proportions are weighted based on the denominator (occupancy), while in regression analysis compositional changes are considered by including the proportional change in respective age- and sex-variables (see methods pages 15, 21, 22 and 27). To account for compositional changes in countries of origin we used the overall number of patients per months and their countries of origin from subset 1 (Sensitivity analysis 1) as denominators due to the lack of data on country of origin in the collected occupancy data.

We have clarified this in the discussion as follows (p.14):

"The strength of our analysis lay in the quasi-experimental situation, in which data was collected before and during the COVID-19 pandemic in a comparable and consistent way by means of unified EHR. We covered a time period of more than 4 years, and adjusted for potential influences on the outcomes related to individual and centre-related aspects, such as changes in size and composition of denominators. All estimates in the main analysis account for increase or change in the denominator in terms of quantity, but also in terms of composition related to age and sex composition. In descriptive analysis, incidence proportions are weighted based on the denominator (occupancy),

while in regression analysis compositional changes are considered by including the proportional change in respective age- and sex-variables.”

We also clarified the role of country of origin, page 15 by rewording a sentence as follows:

“To account for compositional changes in countries of origin we hence used the overall number of patients per months and their countries of origin in each of the 21 centres.”

(2) Were they the same individuals followed up over time, or was it a mixture or was the study unable to tell individuals from another? The text within the main manuscript is not clear in this regard. If it was "hospital count" data then make this clearer.

The study followed-up individuals over time. Each recording of a diagnoses (numerator) is assigned to a single individual, and this individual is “recognized” in any repeat visits and is *not* double-counted for the calculation of incidence proportions. Incident cases are counted and calculated based on unique individual-level data.

the study followed-up individuals over time, which is a key strength of the analysis. Each recording of a diagnoses (numerator) is assigned to a single individual in the EHR, and this individual is “recognized” in any repeat visits and is not double-counted for the calculation of incidence proportions. Incident cases are counted, and incidence proportions calculated based on unique individual-level data distinguishing cases and the respective population at risk attending the clinics. However, estimates using denominators collected from authorities on a monthly basis are aggregated, so that the population at risk for the calculation of incidence proportions with occupancy data may have been over-estimated if individuals remain in the centre for more than a single month, which is very likely. In this case, i.e. in data-subsets 2 to 4 and the main analysis, as well as sensitivity analyses 2 and 4, our estimates are an under-estimation of the true incidence proportion of outcomes, especially in time periods with low in- and out-flows of refugees into the centres, as was the case during the pandemic. Analyses using subset 1 (i.e. sensitivity analysis 1) based on the electronic health record data alone are not affected by this limitation as the denominator used in these analyses is the number of unique patients per month.

Given the acceptable consistency between findings of the main analysis and sensitivity analyses in terms of direction of effects, we believe the potential under-estimation of incidence proportions in occupancy-based analyses does not substantially affect our results regarding the impact of the COVID-19 pandemic on patterns.

We have added this to the discussion nearly verbatim, see page 16 onwards.

(3) The methods could use more detail on these centers and setting – how long are stays ususally in the center? Where do refugees go after the center?

We have added further details on the centers in the methods, including information on duration of stay (page 19), but we need to emphasis that routine data and validated information on duration of stay is not readily available on a national scale:

“Asylum-seekers are obliged to stay in the reception centres for up-to 18 months or longer before they are transferred to district-level accommodation centres. Asylum-seekers from presumed safe countries of origin must stay in the centres until the end of their asylum process, but maximum for 24 months. The duration of stay can hence vary considerably depending on stocks and flows, countries of origin, federal state regulations and processes, and the speed of the asylum process. In Baden-Wuerttemberg, state level authorities report an average duration of stay between 8 to 18 weeks, but underlying data are not public and cannot be validated independently. Other states do not report average durations of stays within centres on a routine basis. Most reception centres provide on-site health care facilities. These are usually equipped to provide basic primary care services within a heterogeneous infrastructure and with a varying quality, while some centres provide additional services ranging from specialist care to psychosocial services”.

What happened to the other 70% of centres in Germany during the pandemic - did these remain open or closed, which could have impacted centre population demographics?

We had already mentioned these aspects in the introduction of the first submission (pages 3-4). We have now additionally elaborated on this aspect in the methods in this revision, see page 19 top:

“Measures in response to the pandemic comprised decongestion, mass quarantine, as well as reduction of health, social, and other service. Although measures were implemented in a heterogenous way across the country, the system of centralized reception remained operational, and reception centres were not closed or shut down in favor of decentralized housing”.

(4) How were indicators chosen and prioritised?

We have added details as follows under methods:

p.20: “Indicators were developed and consented within the consortium by practitioners, healthcare providers, and authorities in a collaborative research approach.”

And p. 24:

“Indicators with very few observations per centre-month (e.g. congenital disorders, multimorbidity) were excluded from the analysis to avoid problems with convergence and fitting of regression models. “

Although it is unlikely, a change in the demographic or more refugee admissions could be an alternative explanation for the increase in certain conditions, this remains a bias in the study design.

We agree that a change in demographics or admissions could be an alternative explanation for the increase or reduction in certain conditions, but we disagree that this remains a bias in the study. As clearly outlined in the manuscript, and as emphasized in the revisions, changing denominators in terms of *both* quantity *and* composition were considered in the main analysis and in the sensitivity analyses 1-3. No additional changes performed.

Parts of the paper are quite difficult to read and follow. Whilst the details provided in the Nature Communications summary on the study design were very clear - I think these pieces of text need to be integrated into the methods section (both in the manuscript and the supplementary section).

We very much appreciate this comment and feedback, and have integrated large sections of the Nature Communications Reporting summary into the manuscript where appropriate. In particular, amendments have been made on the following pages:

p.16: methods, first para

p.18, methods, 2nd and 4th para

p.23, methods, added Sex and gender considerations, Details on Race, Ethnicity and Other Social Categories, Statistical Software and Code

p.24, methods, added Ethics statement

Reviewer #2 (Remarks to the Author):

What are the noteworthy results?

During the pandemic, among refugees there was an increased incidence of injury, violence and mental health conditions. Declined incidence was seen in respiratory conditions.

Thank you for the summary of noteworthy results.

Will the work be of significance to the field and related fields?

It will be of limited significance, as most of the findings include some uncertainty, and the findings are mainly in line with what has been already known.

We agree the findings include some uncertainty, but this is often the case in research in principle and even more true related to populations in precarious settings and for which data situations are far from ideal due to resource constraints or due to political reasons that lead to neglect of such groups in health information systems. We agree the findings are, not completely, but widely, in line with what has been already known which is however an important finding to corroborate and consolidate the existing body of evidence base which is in our view rather thin as far as refugees in camps are concerned.

How does it compare to the established literature?

The results are in line with the established literature.

Does the work support the conclusions and claims, or is additional evidence needed?

The work supports the conclusions and claims.

Are there any flaws in the data analysis, interpretation and conclusions? Do these prohibit publication or require revision?

I did not find any flaws in the data analysis, interpretation or conclusions.

Is the methodology sound?

I find the methodology used sound for this purpose.

Does the work meet the expected standards in your field?

The work meets the expected standards, the results are presented in a clear way and the language is excellent.

Is there enough detail provided in the methods for the work to be reproduced?

Enough details for the methods are presented.

I do not have any specific suggestions for changes or any errors to be corrected.

We thank the review for the above positive comments on our manuscript.

As a general comment, I would stress that using diagnosis codes as the indicator for changes in disease patterns over time during a pandemic has several uncertainties:

- The care seeking patterns were probably influenced.
- The access to care was probably influenced.
- The health care routines and personnel at the centers were probably affected. This might have changed the coding behaviours and the diagnostic criteria.

- No comparison group was used, only temporal changes were analysed. One interesting approach would have been to compare changes in incidence between refugees with the general population.

I suggest that these limitations would be better reflected in the discussion, and if possible some data presented about these factors (number of visits, changes in personnel, access to care, waiting times etc).

We agree with these points, and believe we have outlined how we have tried to address and mitigate some of the uncertainties (see discussion, p.15 onwards). We also agree that comparing patterns among refugees with general population would have been interesting, but there is a lack of data sources allowing for such comparative analyses, which is why we refrained from this approach.

We have however compared and contextualized our findings with external evidence where appropriate and possible, see for example the section in discussion where we compare patterns in declining respiratory conditions with external evidence in resident populations, page 14:

“Our study shows that respiratory conditions considerably declined during the pandemic, mostly likely as a consequence of the strict pandemic control measures implemented in the refugee centres. This finding is in line with international studies in refugee populations³⁶ and national studies in the resident population in Germany³⁷.”

No changes performed.

Reviewer #3 (Remarks to the Author):

I am writing to comment on the research article "The Impact of the COVID-19 Pandemic on Morbidity Patterns in Refugee Accommodation Centers in Germany" which was recently submitted to your journal. While the study provides valuable insights into the health impacts of the pandemic on refugees using a unique natural experiment design, I found the paper extremely difficult to follow due to several issues.

The authors made excessive use of technical terminology without clear definitions. Terms like "zero-inflated negative binomial models", "peri-pandemic", "quasi-experimental situation", and "interrupted time series design" were used without explaining what they mean for non-expert readers. This made it challenging to understand the methodology and analysis.

We very much appreciate this comment, and thank the reviewer for flagging the importance of accessibility of our manuscript to non-technical audience and readership. We adjusted the introduction, and the beginning of the results section, and have explained in plain language the respective terminology (pre-pandemic, peri-pandemic, quasi-experiment, zero-inflated negative binomial models, mixed effects). For interrupted time series we added a reference as this is well explained elsewhere.

The authors found an increased in incidents of injuries, mental disorders, 39 psychotherapeutic drug prescriptions, and genitourinary diseases and a decrease in respiratory diseases decreased.

More demographic details about the refugee population would have helped interpret the findings. Beyond just gender and adult status, providing information on countries of origin may explain why injuries and mental health problems rose so much during the pandemic period. It is also unclear how refugees were permitted to enter Germany during this strict time.

In our Sensitivity Analysis, we considered the potential effects of country of origin and adjusted for this factor, so that we can rule out that this and other sociodemographic factors had substantial impacts on our results. In the Supplementary Material, Table 1, we also present all details on the five most frequent countries of origin in our datasets (Nigeria, Afghanistan, Syria, Iraq, Türkiye) broken down by year.

We have added these details to the beginning of the results section, p.7:

“The five most frequent countries of origin among all refugee patients (N = 215,864) across all centres (2018 – 2023), weighted by the centre’s population size, were Afghanistan (Mean \pm standard deviation: 18 % \pm 18), Iraq (3.2 % \pm 4.8), Nigeria (8.6 % \pm 13), Syria (12% \pm 19) and Türkiye (4,2% \pm 7.6), while percentages broken down by year varied (Supplementary Table S1).”

The authors could have discussed whether the refugee population tended to be older prior to COVID-19 and if age could explain some of the health impacts found.

We have checked the mean age of the patient population before and after the onset of the pandemic (details see methods, page 20). Overall, patients after March 2020 were on average about one year younger. As differences overall and by center appear marginal (See Supplementary file, chapter 3.3., page 108), we kept age as dichotomous variable in the analysis and believe this is sufficient adjustment for the effect of age on potential patterns in incident diagnoses.

It was also surprising to see a decline in respiratory disease reported without further explanation. Including data on the refugee's country of origin and COVID-19 infections rate themselves would have been insightful.

As outlines in the introduction, the methods, and the results, we have adjusted for the most relevant countries of origin, so that variation in respiratory disease incidence is not likely to be due to differences in country compositions.

We disagree that there has been no further explanation related to the decline in respiratory disease and believe this detail may have been overlooked. The section on page 14 of the discussion contextualized this finding as follows:

“Our study shows that respiratory conditions considerably declined during the pandemic, mostly likely as a consequence of the strict pandemic control measures implemented in the refugee centres. This finding is in line with international studies in refugee populations³⁶ and

national studies in the resident population in Germany³⁷. The rising time trend, on the other hand, may be an indication of the subsequent and sequential relaxation of measures over time which gradually individual behaviour change and lower adherence to and enforcement of social distancing or wearing of masks.”

Other patterns have been contextualized along similar lines. No changes performed.

The graphs presented were difficult to interpret without labels on the categories and clearer axis labels.

We are not sure which figures are meant here. All submitted figures are clearly linked to a title, detailed legends, and have axis labels.

Comparing the findings to general population health trends in Germany and globally could provide important context.

Throughout the discussion, we have made comparisons were possible and feasible, but a general comparison to trends in Germany and globally is out of scope and we believe may also not be fully supported by the underlying data as data sources vary substantially. Where appropriate, e.g. comparing surveillance data on respiratory conditions among residents with the patterns found among refugees from this surveillance system, it is already included in the manuscript.

Overall, while the study design was valuable, clearer presentation of methods, definitions, data set details, population details and discussion of implications are needed for readers to fully understand and evaluate results. I had to read it several times to understand and the end of the day, it's hard to judge how the findings for refugees in Germany compare with the general population.

We have added more details on respective sections, also to address comments of other reviewers and believe the paper has no gained further clarity. Comparison to general population: please refer to above comments on this topic.

Reviewer #4 (Remarks to the Author):

The manuscript entitled "Impact of COVID-19 pandemic on incident diagnosis patterns in German refugee centres: quasi-experimental study, 2018-2023" presents unique surveillance data to compare morbidity patterns among refugees in Germany before, during and after the COVID-19 pandemic, covering the timeframe from October 2018 to April 2023. The paper is generally well written and the analyses are rigorously conducted. I see high scientific value in the paper given the scarcity of evidence we have so far on refugee and migrant health. I would generally consider the paper as suitable for Nature Communications.

We thank the reviewer for this very positive comment and agree that evidence is available, but truly scarce.

However, I advise the authors to add further clarity and context to the paper, following the points I outline below.

Major comments:

1. There is some conflation between the terms migrants and refugees. The authors need to specify what population they look at. If I understand correctly, their sample does not include migrants who have resided in Germany for several years but only refugees who are waiting for their asylum cases to be approved? But this is not clearly spelled out and should be explained in more detail.

We have added further details on the study population under methods, page 27 (See section Race, Ethnicity, and other social categories of the study population), which reads as follows:

“Race, Ethnicity and other social categories of the study population

No variables of race or ethnicity were used. Country of origin, i.e. nationality recorded in the medical records system, was used in the regression models to adjust for differences in incident diagnoses that may be attributable to compositional changes of the underlying refugee population in a centre, rather than to the analysed exposure (COVID-19 pandemic). Nationalities included in the analyses were restricted to those countries with the most frequent share among the refugee population across all centres. We use the social category "refugee" in our manuscript and the analysis, acknowledging that this category subsumes a heterogeneous population. We further use this term as "umbrella term" for persons who are registered in the centres and live there with different residence status as asylum claimants, asylum seekers, resettlement refugees, accepted refugees, or tolerated individuals. As our data is based on medical records data, no information was available for the different residence status to allow for more nuanced disaggregation, or in order to evaluate and assess any within-group variations among the social category of “refugees”. The study population did not contain general migrants, but was confined to refugees living in camp-like or institutionalised accommodation facilities (reception centres).”

2. The authors should provide more context on how refugees in Germany are usually integrated into the formal health care system. Is all medical care they receive delivered on-site or is it common for refugees to also attend regular primary care facilities (Hausarztpraxis)? Related to this, it needs to be specified whether we can expect that the PriCareNet likely records the complete medical history of the refugees upon their arrival in Germany or whether some refugees may have used other care services/facilities and would thus not appear in the PeriCareNet records.

We have added details on this important aspect under Methods, in the section “Setting and data sources”, especially on pages 19/20:

Asylum-seekers are obliged to stay in the reception centres for up-to 18 months or longer before they are transferred to district-level accommodation centres. Asylum-seekers from presumed safe countries of origin must stay in the centres until the end of their asylum process, but maximum for 24 months. The duration of stay can hence vary considerably depending on stocks and flows, countries of origin, federal state regulations and processes, and the speed of the asylum process. In Baden-Wuerttemberg, state level authorities report an average duration of stay between 8 to 18 weeks, but underlying source data are not public and cannot be

validated independently. Other states do not report average durations of stays within centres on a routine basis.

Most reception centres provide on-site health care facilities. These are usually equipped to provide basic primary care services within a heterogeneous infrastructure and with a varying quality, while some centres provide additional services ranging from specialist care to psychosocial services^{27,43}.

Measures in response to the pandemic comprised decongestion, mass quarantine, as well as reduction of health, social, and other service. Although measures were implemented in a heterogeneous way across the country^{11,14}, the system of centralized reception remained operational, and reception centres were not closed or shut down in favor of decentralized housing⁵.

And on page 21:

Individuals are included in the medical records if they seek care in one of the on-site medical clinics established for refugees, so that data are "utilisation data", which means that health data of individuals who did *not* seek care is not included. Individuals may also seek care outside of reception centres, but this usually requires approval, and is linked to several geographic, linguistic and administrative barriers, which is why on-site services usually constitute a low-threshold and first anchor point for patients seeking care. In case of serious and chronic conditions, data of referrals or re-referrals to providers of the regular health system are recorded in Ref.Care in the refugee centres as well to ensure ongoing treatment, so that medical history can be regarded as fairly complete. However, it cannot be ruled out that major diagnoses have been not recorded in case of treatment outside of the facilities in the regular health system.

3. While the authors provide detailed information on how COVID policies related to testing, quarantining and mask wearing affected refugees, there is no mentioning of how the German vaccination campaign considered refugees. Were these groups prioritised in the administration of vaccine doses or did they only get later access to vaccination?

Refugees were prioritized in policies but this was only "on paper" and not effectively implemented. In fact, refugees were in practice de-prioritised and received vaccinations rather late and after groups categorized by the national public health agency as having a lower priority. There are no formal and systematic assessments, but media reports evaluating and assessing the roll out of the vaccination campaigns. We have added this in the discussion and referenced to respective media reports which also refer to field insights of the author group of this manuscript.

p.14:

"The rising time trend may also be an indication of the rather slow, and late implementation of effective vaccination measures against COVID-19. Media reports indicate and prove that despite the prioritisation by the national public health agency of refugees in the national vaccination policy, local implementation was ineffective and de-prioritised refugees in practice leading to delayed vaccination up-to April 2021³⁸. As formal evaluations of vaccination rates in refugee populations are scarce globally¹⁵, no further insights exist to validate or assess the impact of potentially delayed vaccination roll-out in this group."

4. The dataset only includes information from refugee centers in three German states (Bundesländer). The authors need to more explicitly acknowledge this upfront (i.e. in the abstract or introduction), given that the presented data can therefore not be considered as representative for Germany as a whole.

We have added this upfront in Abstract and Introduction. We do however not fully agree that the sample is not comparable to the refugee population in Germany as a whole, see answer to next comment.

Relatedly, the authors should also extend their limitation section by a discussion of the external validity of their results (to the other 13 German states), especially considering that the data they present is from West Germany and that refugees in East Germany may even face higher disadvantages due to both lower capacity of the healthcare system in Eastern German states as well as higher levels of racism/hostility towards refugees.

Distribution to the states in general, i.e. also to the other 13 states, occurs in a quasi-random manner based on administrative quota which means that geography alone is not in itself a factor limiting external validity. We have added details on this on page 15:

“As for the risk of compositional effects on our results, the sample is widely representative with respect to age, sex, and distribution of nationalities to the general refugee population in Germany. The average age was about a year lower in the refugee patients after onset of the pandemic, and overall differences broken down by centres (Supplementary File) appear marginal which makes age differences unlikely as explanation for the changes in incident diagnoses. Furthermore, allocation to states, including the three states under study, occurs on a quasi-random manner based on administrative quota, which is why it is very likely that there is no systematic difference refugee characteristics or underlying morbidity between different states. We minimised the risk for residual confounding by adjusting for sex, age, and nationality (amongst others) in our regression models. While reception centres and states may differ with respect to contexts and pandemic measures, we considered this level as random effect in the analysis.”

We agree however, that unmeasured confounding may exist due other factors related to geography, e.g. political hostility in Eastern Germany. We have added this aspect in the discussion following the above section, p. 16:

“While the risk for compositional bias or residual confounding by compositional factors is low due to above quota and our adjustments, contextual factors in other regions, e.g. political hostility against refugees in Eastern Germany³⁹ may have led to even higher disadvantages for refugees due to higher levels of racism/hostility towards refugees³⁹, as well as due to the lower capacity of the regional healthcare system.”

5. The analysis includes aggregated information on characteristics of the included refugee centers, including occupant numbers, composition of adult and child refugees, and composition of male and female refugees. I believe that it would add great value if the authors tried to add additional information on the included refugee centers, including (1) proxies for overcrowding/inadequate shelter (i.e., squaremeter per occupant or number of rooms per occupants?), (2) number of staff working in the center/ratio staff:occupants, and possibly (3) annual per capita budget/funding available to the refugee center.

Unfortunately, such data are not readily available, and if they exist, they are fragmented (i.e. they are not available for all centres and time points), and not captured in a standardized way. Authorities partially refrain from providing such data, so that we cannot include this relevant information on contexts in the analysis. We have addressed potential variations due to “contextual factors” in our analytical approach (centres as random intercept), however, this approach “adjusts away” potential centre-related differences and does not allow to

assess which centre characteristics in fact made a difference. No changes performed.

6. The authors should elaborate more in detail how the data is entered into the PeriCareNet database. Who enters the data? How often is data entered? How accurate and complete is the entered data?

We have added this briefly in the methods, page 20:

Ref.Care is used by health professionals, who are the data holders of the individual-level patient data in on-site health care facilities. They enter the medical data into the EHR as part of their routine clinical work, so that no parallel data entry effort is required. Data entry fields are both standardized and allow for free texts (e.g. for medical history), but all data used in this analysis is based on standardized codes and diagnoses (see below).

Data on accuracy and completeness is provided on page 20. Differences in coding behavior are discussed in the discussion and measures provided to account for this are incorporated in the analysis (random intercept of centres).

7. On p. 8, the authors first point to the lack of psychosocial care services in most reception centers (line 228) but then observe an increase in prescriptions of psychotherapeutic drugs (line 231). There needs to be a more detailed explanation of which psychiatric and counselling services refugees in Germany usually have access to (and how/which kind of these services may have been constrained/disrupted due to the pandemic).

As described in the introduction, the services are very heterogenous so it remains hard if not impossible to make a generalized statement on this for the three federal states. The prescriptions can also be made by GPs in absence of for example psychiatric services on-site. We have hence performed no changes as there is no further “valid” information we could provide on this, and any further elaborations on services on-site would be speculations, which we would like to avoid.

8. The authors observe an increase in genital and other infections during the pandemic and largely interpret this as a consequence of COVID infections. However, I was wondering whether the pandemic may also have caused changes in sexual behaviours, greater exposure to sexual violence, and disrupted access to sexual health care services, which may serve as an alternative explanation for the observed increase. See for example:

<https://link.springer.com/article/10.1186/s12889-022-13866-7>

<https://www.tandfonline.com/doi/full/10.1080/26410397.2020.1822493>

<https://gh.bmj.com/content/7/10/e009594.abstract>

We appreciate this comment and agree these are relevant pathways that could potentially explain the observed patterns. We have included a section in the discussion as follows, p.13.

“The rise in diseases of the genitourinary system observed in this refugee population may also be explained by potential changes in sexual behaviour, including higher rates of transactional sex³⁶ and gender-based violence³⁷, and/or changes in access to sexual and reproductive health services outside of the facilities³⁸. Further research is, however, required to confirm potential pathways.”

9. The authors treat the peri-pandemic phase as fully homogeneous as their analysis only differentiates - following a binary logic - between pre-pandemic (0) and post-pandemic months. However, the timeframe from the onset of the COVID-19 pandemic until April 2023 was very heterogeneous in terms of COVID-19 infection numbers, mortality rates and - linked to this - policy measures such as lockdowns or contact restrictions that were adopted to contain the spread of the virus. Would it make sense to include a more fine-grained analysis (at least as sensitivity checks) that tries to model the "strict" and "loose" phases of the pandemic (e.g. drawing on Google mobility data) in a more nuanced way? This could also help establish whether increases in certain health outcomes (e.g. mental health diagnoses) are more substantial during/after times with tougher restrictions.

We very much appreciate this thoughtful comment. We have now implemented additional sensitivity analyses with the attempt to capture and evaluate nuances during the peri-pandemic phase beyond a dichotomous approach. While the suggestions of the reviewer with respect to strictness based on mobility data is not feasible in our context, we have implemented a more fine-grained approach as follows:

In Sensitivity analysis 4, we have implemented an analysis that distinguishes three phases of the pandemic:

- Pre-pandemic phase (before 02/2020)
- Phase 1 of the pandemic: representing the early phase until the onset of vaccine availability, i.e. 02/2020 – 07/2021
- Phase 2 of the pandemic: Emerging variants phase (e.g. delta, omicron etc.) and increasing vaccine role out, i.e. 08/2021 to 06/2023

The results are presented in Supplementary File, Chapter 5 (Figure S26), and they show that our results are widely robust to the nuances of different pandemic phases, as the observed patterns for e.g. disability, consequences of external causes of injuries, circulatory conditions, and psychological conditions remain unaffected and are consistent to patterns in our main analysis. Estimates for respiratory conditions are also consistent with previous main and sensitivity analysis, showing tendencies to declining incidence in the early phases of the pandemic, and rising trends in later phases.

In Sensitivity analysis 5, we have implemented an analysis to examine seasonality effects, comparing effects of spring, autumn, and winter periods with summer periods (2018-2023), respectively. The results are presented in Supplementary File, Chapter 5 (Figure S27), showing that seasonality effects exist in particular for consequences of external causes of injuries, with higher incidences in summer periods compared to spring, winter and autumn, respectively. This may be due to route dependent effects (more dangerous routes during summer), heat related effects on violence, or higher probability of injuries related to outdoor activities. The incidence of psychological conditions was higher in winter periods (compared to summer periods), while respiratory conditions were significantly higher in winter and autumn (compared to summer periods). No considerable seasonality effects were observed for the remainder of diagnoses. The patterns prove the high plausibility of the surveillance data, and are overall consistent with the patterns observed in the main analysis.

We have added these details to the methods section (p.24-25) and result section (p. 10-11).

Minor comments:

1. Introduction: The introduction is well written and nicely motivates from a health equity perspective why it is important to focus on migrant and refugee health and why refugees might be particularly vulnerable to facing adverse consequences due to the pandemic. I suggest that the authors extend the second paragraph (p.3) to also note that migrant/refugee populations may face health disadvantages during pandemics due to their disintegration into the regular health care system and language barriers/health literacy barriers that may hamper their access to necessary health information.

We have added this to the 2nd para of the introduction as follows:

“They may face additional health disadvantages during pandemics due to their disintegration into the regular health care system, as well as language barriers that may hamper their access to necessary health information”.

2. Introduction, paragraph 3: "... and quarantine upon transfer" - the authors should specify what kind of transfer they refer to here. Is this the transfer from one refugee center to another?

We have specified this.

3. Discussion, p. 8, line 220 - the authors attribute the rise in injuries partly to experiences during flight, however, this would be a factor independent of the COVID-19 pandemic and would have occurred before the patients joined the refugee center. Why would violence during the flight have increased in consequence of the pandemic? This is unclear.

We believe this is a misunderstanding as we did not attempt to attribute the rise exclusively to factors during flight. It may also be attributable to factors during the stay in centres, which may have changed before versus during the pandemic. We have now pacified the different possible explanations as follows in the 2nd para of the discussion:

“The 88% rise in diagnoses related to injury and consequences of external causes may indicate an increased experience of violence before or during flight, e.g. due to the onset of the Russian war against Ukraine. However, this is unlikely as we adjusted for compositional effects, and because major nationalities among refugees remained widely constant 2018-2023, and Ukrainian nationals (whose numbers peaked from 2022 onwards) mostly resided in private accommodation outside reception centres. The results may also indicate higher violence *during* refugees’ stay in the centres. Systematic reviews of media reports found that violent incidences in refugee centres during COVID-19 outbreaks were common. These were related to social tensions within the centres or to enforcement of the strict pandemic rules which partially required, or were related to, police operations¹⁰.”

4. Discussion, p. 8, line 225: "... and other adverse experiences" - please specify these.

We have specified this:

“Previous research also showed that refugees living in reception centres are at a higher risk of experiencing mental health disorders due to crowded living conditions, reduced autonomy, lack of privacy and other adverse experiences often associated with life in such centres, such as uncertainty of asylum processes and legal status, geographical remoteness, and social isolation²²⁻²⁶”

5. The authors should possibly give some examples of what "diseases of the genitourinary system" could be since this is likely not common knowledge among most readers.

We have specified a few examples in the discussion:

“Our analysis also showed a 51% increase in “Diseases of the genitourinary system” (N00-N99) following the onset of the COVID-19 pandemic, which includes, for example, urinary tract infections, kidney disease, and inflammatory or non-inflammatory diseases of male and female genital organs.”

POINT BY POINT RESPONSE TO REVIEWERS

Reviewer #1 (Remarks to the Author):

Thanks for addressing my comments. I have no additional comments.

Thank you, we are glad that our revision addressed all queries and comments.

Reviewer #4 (Remarks to the Author):

I thank the authors for the very thorough revision of their manuscript and the additional sensitivity analyses conducted, which all confirm the robustness of the results. All my comments have been adequately addressed and the responses have been well explained. I just have one very minor remaining comment. The authors write: "Nationalities included in the analyses were restricted to those countries with the most frequent share among the refugee population across all centres." - could the authors list these nationalities? I believe it's likely Iraq, Afghanistan, Syria, Ukraine, Somalia...? What else? Other than this, I really like the paper and am looking forward to seeing it published soon.

We have added the details on nationalities that went into the analysis and the regression models. The sentence reads as follows:

Nationalities included in the analyses were restricted to those countries with the most frequent share among the refugee population across all centres (Afghanistan, Iraq, Nigeria, Syria, Turkiye).